# Context-aware deep learning enables high-efficacy localization of high concentration microbubbles for super-resolution ultrasound localization microscopy

YiRang Shin[1,2], Matthew R. Lowerison[1,2], Yike Wang [1,2], Xi Chen[1,2], Qi You[1,3], Zhijie Dong [1,2], Mark A. Anastasio [1,2,3,4] & Pengfei Song [1,2,3,4,5] ✉

Ultrasound localization microscopy (ULM) enables deep tissue microvascular imaging by localizing and tracking intravenously injected microbubbles circulating in the bloodstream. However, conventional localization techniques require spatially isolated microbubbles, resulting in prolonged imaging time to obtain detailed microvascular maps. Here, we introduce LOcalization with Context Awareness (LOCA)-ULM, a deep learning-based microbubble simulation and localization pipeline designed to enhance localization performance in high microbubble concentrations. In silico, LOCA-ULM enhanced microbubble detection accuracy to 97.8% and reduced the missing rate to 23.8%, outperforming conventional and deep learning-based localization methods up to 17.4% in accuracy and 37.6% in missing rate reduction. In in vivo rat brain imaging, LOCA-ULM revealed dense cerebrovascular networks and spatially adjacent microvessels undetected by conventional ULM. We further demonstrate the superior localization performance of LOCA-ULM in functional ULM (fULM) where LOCA-ULM significantly increased the functional imaging sensitivity of fULM to hemodynamic responses invoked by whisker stimulations in the rat brain.

Single-molecule localization microscopy (SMLM) is an established super-resolution optical imaging technology that uses the stochastic blinking of fluorophore emissions within a dense sample[1,2]. By localizing individual emissions and accumulating the localized positions, SMLM reconstructs a super-resolved image, offering an order-of-magnitude improvement in imaging spatial resolution[3]. The concept of localization microscopy has been successfully adopted by the ultrasound community to overcome the acoustic diffraction limit. As an analog to SMLM, ultrasound localization microscopy (ULM) uses ultrasound contrast agents (i.e., microbubbles or MBs) that flow within the blood vessels as

individual point targets to achieve super-resolution[4,5]. The precise localization of each MB increases the ultrasound imaging spatial resolution by an approximate factor of ten[6]. When combined with deep penetration of ultrasonic waves, the accurate localization of MBs offers the potential for reconstructing the deep vascular network with micron-scale spatial resolution. This key advantage renders ULM a powerful tool for noninvasive probing of deep tissue microvasculature in numerous preclinical and clinical applications[7].

As with all imaging techniques, ULM is not without limitations. At present, a key limitation of ULM is the long data acquisition time,

[1]Beckman Institute for Advanced Science and Technology, University of Illinois Urbana–Champaign, Urbana, IL, USA. [2]Department of Electrical and Computer Engineering, University of Illinois Urbana–Champaign, Urbana, IL, USA. [3]Department of Bioengineering, University of Illinois Urbana–Champaign, Urbana, IL, USA. [4]Carle Illinois College of Medicine, University of Illinois Urbana-Champaign, Urbana, IL, USA. [5]Neuroscience Program, University of Illinois Urbana-Champaign, Urbana, IL, USA. ✉e-mail: songp@illinois.edu

which arises from the inherent trade-off between MB concentration and MB localization efficiency and accuracy. To achieve precise MB localization, ULM requires a limited number of MBs per imaging frame through low MB concentrations to ensure that the MB signals are spatially separate and localizable. However, a lower MB concentration also means a slower accumulation of adequate MB localizations to populate the vessel lumen, which can take several to tens of minutes[8,9]. In contrast, a higher MB concentration accelerates the MB localization filling process in theory. However, in practice, they also increase the likelihood of MB overlap, complicating the separation of adjacent MBs and reducing overall localization efficiency. Therefore, a higher MB concentration does not necessarily translate to faster ULM imaging. As such, enhancing MB localization efficiency with high MB concentrations is a challenging yet essential task to improve the imaging speed (i.e., temporal resolution) of ULM.

Various methods have been proposed to improve MB localization under high-density MB conditions. Earlier studies focused on the use of Fourier-based filters, which separate overlapping MBs into subgroups by leveraging the diverse spatiotemporal flow characteristics of MBs[10]. Algorithms based on sparse image recovery and compressed sensing have also been proposed for localization and super-resolution, assuming a sparse distribution of MBs in each imaging frame[11–13]. However, these methods involve a time-consuming iterative procedure and rely on the construction of an accurate PSF model. Moreover, the effectiveness is limited in regions of high MB concentration where the assumption of sparsity does not necessarily hold true.

Deep learning has emerged as a promising solution for robust MB localization under high MB concentrations. However, the absence of ground truth MB locations in vivo remains a major hurdle for network training for localization tasks. Existing ultrasound modeling and simulation software, such as bivariate Gaussian models[14], Field II simulations[15], and SIMUS[16] have been used to simulate MB signals. However, these simulations fall short of creating complex and spatiotemporally varying MB signals observed in vivo, which can be attributed to numerous factors involved in the MB imaging process. These factors include transmit and receive beamforming, acoustic wave propagation (both linear and nonlinear propagation, attenuation, reverberation, multi-scattering, aberration, etc.[17]), the imaging system (e.g., hardware components such as transducers and system circuitry), imaging settings (e.g., TGC), MB acoustic responses[18], tissue properties (e.g., different types of tissues and blood flow conditions), and the postprocessing components (e.g., clutter filtering, denoising, etc.). To account for these effects, a simulation framework was designed to model nonlinear wave propagation and MB responses, generating single-channel RF signals[19]. A 1D CNN was trained to deconvolve MB signals from raw RF data, enhancing the axial resolution of beamformed images tenfold over standard B-mode imaging. However, this technique requires precise parameter tuning and system characterization to generate realistic training datasets. Any mismatch between simulated and true MB signals can introduce biases into the network's outputs, compromising the localization accuracy.

In this study, we introduce a deep learning-based MB localization technique named LOcalization with Context Awareness (LOCA)-ULM, designed to achieve high-accuracy MB localizations even at high MB concentrations. One main contribution of LOCA-ULM is overcoming the difficulties of generating realistic MB training data for developing deep learning-based localization techniques for in vivo contrast-enhanced ultrasound imaging applications. The proposed generative adversarial network (GAN)[20]-based architecture was able to learn the distribution of real MB signals acquired in vivo and generate diverse MB templates that are essential for developing generalizable solutions for a variety of imaging conditions with different types of biological tissues. To further minimize the mismatch between the simulation and real data, we incorporated ultrasound system noise modeling and key

MB attributes (e.g., brightness levels, lifetime, and movement velocity) into the simulation process. Collectively, LOCA-ULM demonstrated marked improvement in MB detection accuracy and presented a practical solution to solving the domain discrepancy problem when developing deep learning-based imaging techniques that involve the use of MBs.

The second aim of LOCA-ULM is to enhance MB localization performance at higher MB concentrations. Deep learning offers the potential to identify MB centers under imaging conditions and MB concentrations challenging for conventional methods. Approaches such as Deep-ULM[14] and modified subpixel neural network (mSPCN)[21] employ mean-squared error-based regression to transform input contrast-enhanced ultrasound images to super-resolved images. Despite their advances, such image-to-image translation networks require an additional localization step to determine MB coordinates, often involving the detection of local maxima or centroids. In contrast, ULM-GAN[22] circumvents the conventional localization-and-tracking approach by using a training strategy that directly maps the temporal average of short-accumulation ultrasound images to ULM images accumulated from long data acquisitions. Additionally, the use of spatiotemporal 3D-convolutional neural networks (3D-CNN) enables the direct extraction of dense MB networks at high MB concentrations[23]. However, these techniques have thus far been limited to spatial reconstructions of microvasculature, which omits potentially critical physiological biomarkers such as blood flow velocity.

In this work, we adopted the concept of Deep Context-Dependent (DECODE)[24] neural network, which utilizes a joint count loss and localization loss to predict both the probability of emitter detection and its subpixel location. Our work also expands on the application of spatiotemporal networks, such as 3D-CNNs and long short-term memory networks in super-resolution imaging. These networks have been successfully utilized in ultrasound MB imaging for tasks like phase aberration correction[16], localization-free microvessel velocimetry[25], and MB track reconstruction[23]. The DECODE network aims to enhance MB localization performance in a high-density MB regime by integrating spatiotemporal information across adjacent frames. Benchmarking of MB localization was first conducted on simulated datasets, comparing LOCA-ULM to both conventional and existing deep learning-based MB localization techniques. We further assessed the performance of LOCA-ULM using various in vivo imaging models, including different MB concentration levels and state-of-the-art MB separation techniques[10]. Finally, the application of LOCA-ULM to functional ULM (fULM) demonstrated that our method enhances the sensitivity of detecting MB count fluctuations that are correlated with neural activity.

## Results

Figure 1a illustrates the simulation pipeline designed to generate realistic MB images that are used as labeled training data. The simulation is based on a least-squares generative adversarial network (LSGAN)[26] (Fig. 1a "G") which produces synthetic MB templates that resemble those observed in vivo (Methods). LSGAN was initially trained on MB signals extracted from in vivo ultrasound images using a conventional localization algorithm based on normalized cross-correlation (NCC)[27]. Once trained, the LSGAN was used to generate a diverse and realistic set of MB templates that were stored in a bank of MBs (i.e., a collection of MB templates later used for training the DECODE). To create the ground truth MB positions, a list of sub-wavelength MB positions was generated and assigned MB attributes such as MB brightness levels, lifetime, and velocity to emulate real MB flow (Methods). The ground truth positions were then convolved with the synthesized MB templates randomly selected from the bank of MBs, creating realistic simulated images with known ground truths. A representative simulated image using an LSGAN-generated MB signal is shown in Fig. 1b

(LSGAN), which closely resembles real in vivo chicken embryo chorioallantoic membrane (CAM) images (Fig. 1d, real image). In contrast, conventional simulation techniques such as 2D Gaussian modeling and Field II show a mismatch between the simulated and real ultrasound images (Fig. 1b Gaussian and Field II). LSGAN offers the advantage of generating a diverse set of realistic MB templates, including those not observed during training. This extensive collection acts as a form of data augmentation, enhancing the robustness of the network for the task of MB localization. Finally, background ultrasound noise was modeled (Fig. 1c) and added to the simulated MB images to create the final training datasets for the DECODE network (Fig. 1d, simulated image) (Methods).

To address the challenge of localizing spatially overlapping MBs commonly observed at high MB concentrations, we adopted the principles of DECODE and translated DECODE cost functions. The cost function includes emitter count loss and localization loss, and the network is trained for the tasks of estimating MB counts, MB detection probability, MB positions, and MB brightness levels[24]. The count loss optimizes the per-pixel detection probability map in the context of aligning the predicted number of MBs with the true number of MBs. This enables the network to output an MB detection probability map

that highlights the pixels where the likelihood of finding an MB is high. The localization loss is jointly optimized with the count loss to maximize the likelihood of estimating the true coordinates and brightness levels of each detected MB. A Gaussian distribution is modeled for each coordinate weighted by the predicted detection probability, where the mean represents the subpixel location of MBs (Methods). This is more robust than directly using super-resolved images as the network's output (e.g., 1's for the center of MBs and 0 for otherwise), which is difficult to train due to its sparse nature[14,28]. The DECODE network architecture also leverages the spatiotemporal coherence of MB signals by using MB information across successive frames. In the inference stage (Fig. 1e), the network estimates MB locations and brightness levels with in vivo contrast-enhanced ultrasound images as input.

## Simulation study

We first validated the proposed LOCA-ULM localization pipeline using simulation data. We created a test dataset distinct from the training dataset through the simulation pipeline using MB signals extracted from an in vivo CAM experiment (Methods). Five hundred imaging frames with an image size of 100 pixels × 100 pixels (12.3 μm pixel size) were generated for different MB concentrations. The simulated

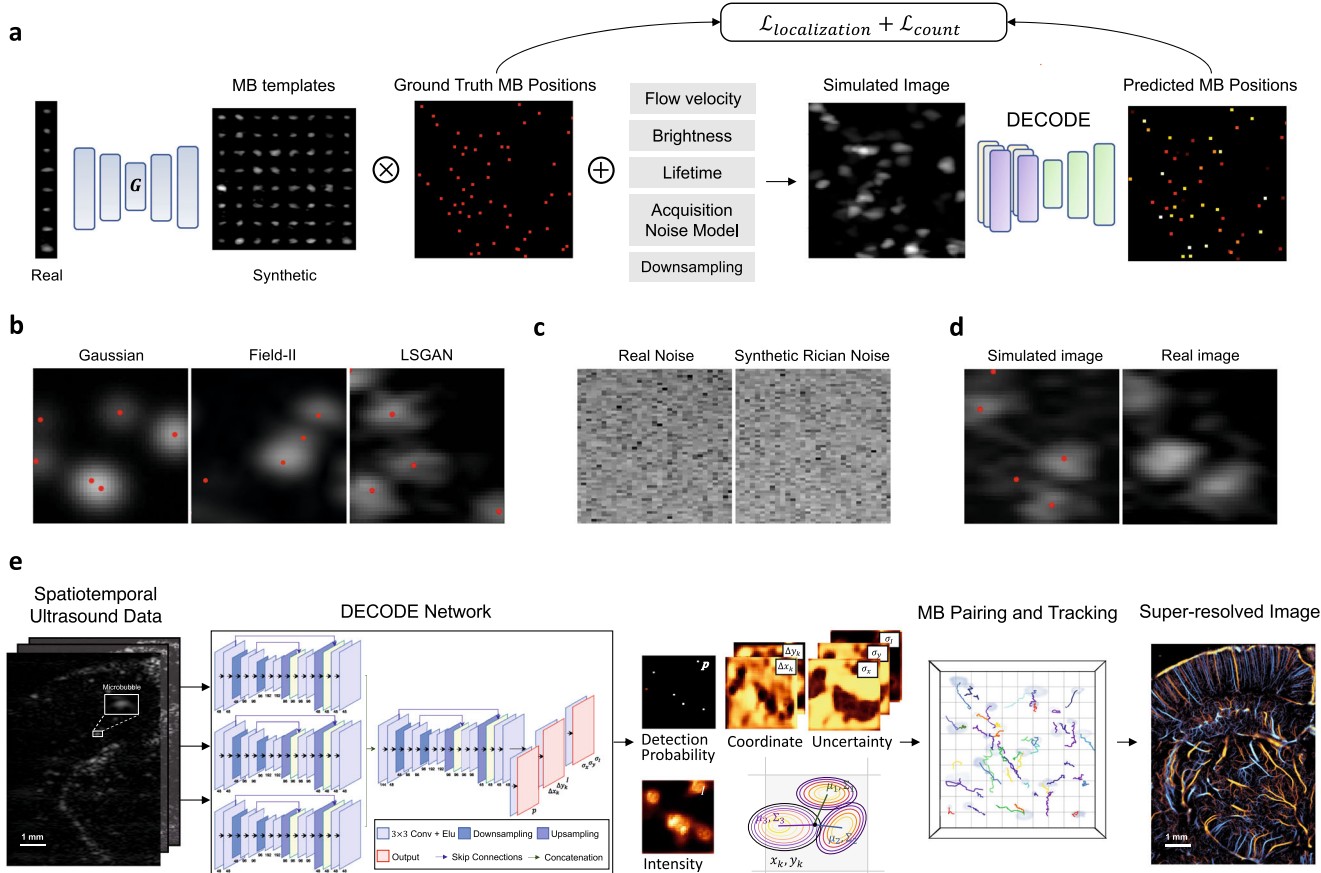

**Fig. 1 | Overview of the proposed LOCA-ULM MB localization pipeline.** LOCA-ULM is a simulation-based supervised learning method using MB templates generated by Least-squares Generative Adversarial Network (LSGAN)[26] and DECODE localization network[24] (Methods -- DECODE Architecture) **a** The LSGAN (*G*) was trained on a large set of MB signals identified by the conventional normalized cross-correlation (NCC) localization algorithm. The LSGAN learns the distribution of real MB signals and generates diverse and realistic synthetic MB templates. The LSGAN-generated MB templates are convolved with simulated ground truth MB positions assigned with MB-specific characteristics (e.g., brightness levels, velocity, lifetime) to create simulated images that closely resemble real data. The simulated images were used to train the DECODE network for localization. **b** Examples of simulated

MB data using different MB modeling methods (Gaussian, Field II, and LSGAN). Red dots indicate the ground truth MB location. **c** Examples of experimentally acquired electrical noise from the ultrasound system, synthesized Rician noise using the proposed method (Methods). **d** Simulated image using LSGAN-based MB templates with added Rician noise and real MB image extracted from the in vivo CAM dataset. **e** DECODE-based ultrasound localization microscopy pipeline. Inference was performed by using in vivo ultrasound data. 2D-DECODE outputs the probability of detecting an MB near pixel $k$ ($p_k$), sub-wavelength spatial coordinates ($\Delta x_k, \Delta y_k$) with respect to the center of the pixel $k$, MB brightness ($I$), and corresponding uncertainties ($\sigma_x, \sigma_y, \sigma_I$). MB pairing and tracking were applied to predicted coordinates, and the final super-resolved ULM images were generated.

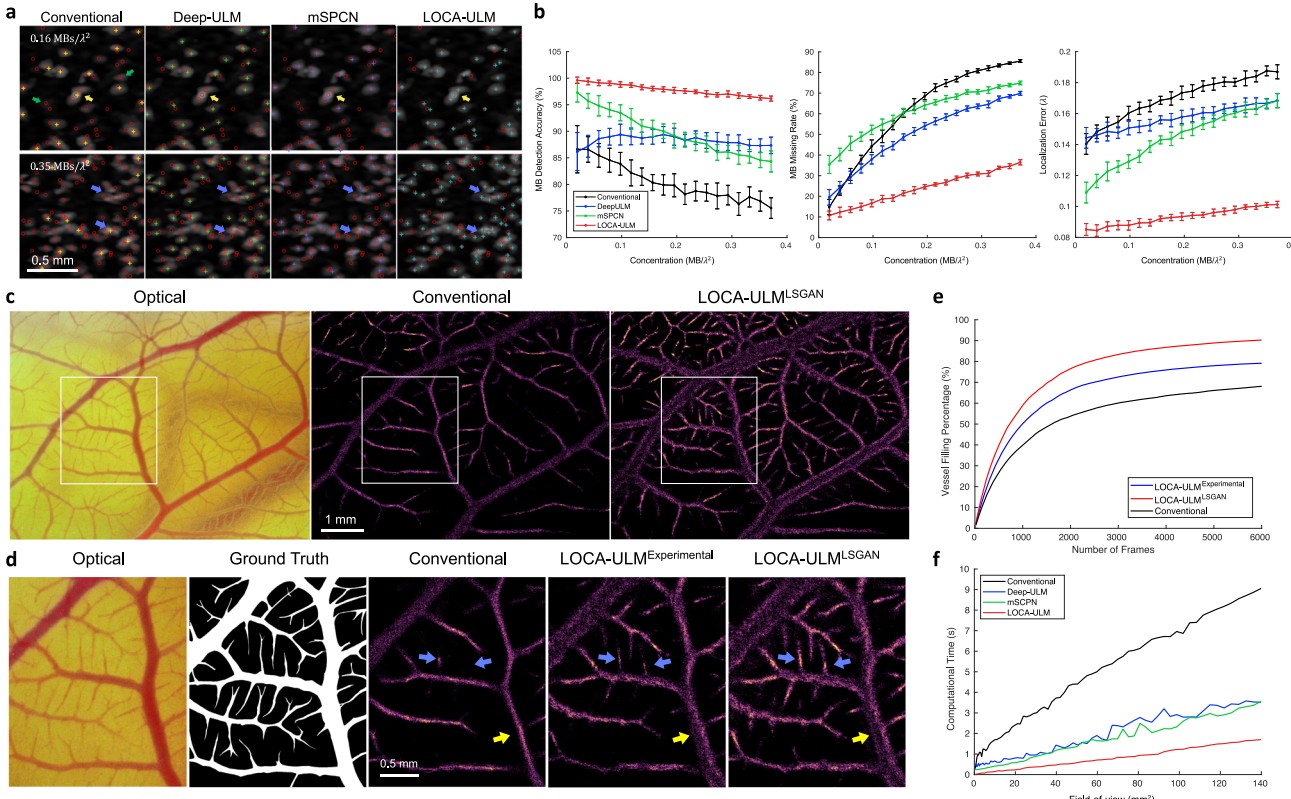

**Fig. 2 | Results of the simulation study and in vivo chicken embryo CAM imaging study. a** Simulation results of conventional localization, Deep-ULM[14], mSPCN[21], and LOCA-ULM with low (0.16 MBs/$\lambda^2$) and high (0.35 MBs/$\lambda^2$) MB concentrations. Ground truth MB positions are marked by red circles, conventional localization by yellow ×, Deep-ULM by green ×, mSPCN by blue ×, and LOCA-ULM by cyan ×. **b** Comparison between conventional localization, Deep-ULM, mSPCN, and LOCA-ULM was performed on simulated frames at increasing MB concentrations, using three performance metrics: MB detection accuracy, MB missing rate, and localization error. The error bars indicate the standard deviation of inference over 500 frames. **c**–**e** Comparisons among conventional localization, LOCA-ULM[Experimental], and LOCA-ULM[LSGAN] MB localization in in vivo CAM imaging. **c** Optical microscopy image of the CAM surface microvessel, along with MB localization images reconstructed by conventional localization and LOCA-ULM[LSGAN]. $n = 1$ experiment. **d** The ROI selected from the optical image and the corresponding ground truth vessel segmentation. Magnified view of the MB localization images marked by the white ROI for conventional localization, LOCA-ULM[Experimental], and LOCA-ULM[LSGAN]. **e** The vessel filling (VF) a percentage of conventional localization, LOCA-ULM[Experimental], and LOCA-ULM[LSGAN], as a function of the number of frames (Methods). **f** Comparison of computational times for MB localization across varying FOV sizes using conventional ULM, Deep-ULM, mSPCN, and LOCA-ULM. Computational times encompass the MB localization process, which includes normalized cross-correlation and regional maximum search for conventional ULM, a network forward pass for LOCA-ULM, and a network forward pass with centroid calculation for both Deep-ULM and mSPCN. Source data are provided as a Source Data file.

average MB concentration ranged from a low density of 0.02MBs/$\lambda^2$ to a high density of 0.37MBs/$\lambda^2$, increasing in steps of 0.02MBs/$\lambda^2$, where $\lambda$ is the wavelength of the ultrasound pulse used for imaging (Supplementary Table 1). Both conventional and deep learning-based localization techniques were used to benchmark LOCA-ULM performance. For conventional localization, we employed the NCC-based method using a pre-defined bivariate Gaussian distribution[27] (Methods). For deep learning-based methods, we compared Deep-ULM and mSPCN (Supplementary Methods). Both Deep-ULM and mSPCN were trained using the same MB simulation pipeline as LOCA-ULM. Unlike LOCA-ULM, which directly outputs MB locations (i.e., MB coordinates), Deep-ULM and mSPCN generate super-resolved MB images where subsequent localization is still necessary.

Figure 2a presents examples of MB localization results using conventional ULM, Deep-ULM, mSPCN, and LOCA-ULM on the simulation datasets with varying MB concentrations. At a low MB concentration (0.16MBs/$\lambda^2$), the MB centers localized by Deep-ULM, mSCPN, and LOCA-ULM aligned well with the ground truth. However, conventional localization had limitations in accurately localizing small and irregularly shaped MBs (green arrows in Fig. 2a), which is likely due to the constraints imposed by convolving a pre-defined Gaussian distribution with fixed shapes and sizes. Importantly, both conventional localization and deep learning-based approaches struggled with

overlapping MBs, which were accurately localized by LOCA-ULM (yellow arrow in Fig. 2a). The disparity in performance increased at high concentration (0.35MBs/$\lambda^2$), where conventional ULM, Deep-ULM, and mSPCN all showed a marked decline in MB detection rate—especially in clustered MB areas (blue arrows in Fig. 2a). In contrast, LOCA-ULM maintained a low missing rate and high localization accuracy even for highly overlapped MBs with various shapes and brightness levels.

The MB localization performance on simulation data was evaluated quantitatively using three metrics: MB detection accuracy, MB missing rate, and MB localization error (Methods). Figure 2b presents a comparison of the performance of LOCA-ULM against conventional localization, Deep-ULM, and mSPCN with respect to increasing MB concentrations. LOCA-ULM consistently outperformed all competing algorithms in MB detection accuracy and MB missing rate, particularly under conditions of high MB concentrations: the average MB detection accuracy of LOCA-ULM was 97.8%, which is significantly higher than Deep-ULM (88.2%), mSPCN (89.7%), and conventional localization (80.4%); the average MB missing rate of LOCA-ULM was 23.8%, which is over a twofold improvement over Deep-ULM (50.3%), mSPCN (60.9%), and conventional localization (61.4%). The enhanced performance of MB localization provided by LOCA-ULM is essential for shortening the data acquisition time for ULM because it allows higher concentration

MBs to be administered in vivo while maintaining a robust MB localization performance with high efficacy.

Figure 2b also shows that LOCA-ULM consistently reduced the MB localization error across all concentrations when compared to conventional localization, Deep-ULM, and mSPCN. The theoretical resolution limit of ULM (i.e., localization error) can be estimated using the Cramér-Rao lower bound (CRLB)[29], which gives the highest resolution of 3.29 μm with the CAM study acquisition settings (Methods). In low-density conditions, LOCA-ULM achieved a localization resolution of 6.47 μm, which is the closest to the CRLB prediction (10.78 μm for conventional localization, 11.09 μm for Deep-ULM, and 8.39 μm for mSPCN).

## GAN-generated MB signals improved LOCA-ULM performance for MB localization in the in vivo CAM imaging study

To demonstrate the effectiveness of LSGAN-generated MBs, we trained the LOCA-ULM network using two distinct simulation datasets. The first, referred to as LOCA-ULM$^{Experimental}$, was trained using MB signals directly extracted from the in vivo CAM data. The second, LOCA-ULM$^{LSGAN}$, was trained with synthetic MB templates generated by the LSGAN network trained on the same set of MB signals used to train the LOCA-ULM$^{Experimental}$. Figure 2e summarizes the vessel filling (VF) percentage for all the localization methods, which includes conventional localization, LOCA-ULM$^{Experimental}$, and LOCA-ULM$^{LSGAN}$ (Methods). LOCA-ULM$^{Experimental}$ and LOCA-ULM$^{LSGAN}$ achieved consistently higher VF percentage and a faster vessel saturation rate than conventional localization. At 6000 frames (total 6 s of acquisition), LOCA-ULM$^{LSGAN}$ reached the highest VF percentage (90.3%), followed by LOCA-ULM$^{Experimental}$ (79.2%), and conventional localization (68.1%). Notably, the VF percentage of conventional localization with respect to the optical image started to plateau around 70%, while LOCA-ULM$^{LSGAN}$ did not plateau until 90%. This result aligns with the observation of under-filling in major vessels using conventional localization, as indicated by the yellow arrows in Fig. 2d. Large vessels are more prone to underfilling with ULM due to higher MB concentration from increased flow rates[8], posing a challenge for conventional methods that often lead to incomplete reconstructions. In contrast, due to the robust performance of LOCA-ULM in high MB densities, both LOCA-ULM$^{LSGAN}$ and LOCA-ULM$^{Experimental}$ filled the large vessels more completely and the size of the vessel was closer to the reference based on optical microscopy (Fig. 2d). Both LOCA-ULM$^{LSGAN}$ and LOCA-ULM$^{Experimental}$ outperformed conventional localization in imaging small vessels as well, with LOCA-ULM$^{LSGAN}$ outperforming LOCA-ULM$^{Experimental}$ as evidenced by the reconstructed microvessels that were less developed in conventional and LOCA-ULM$^{Experimental}$ (Fig. 2d, blue arrows). The findings suggest that synthetic MB signals generated by LSGAN act as a form of data augmentation, allowing LOCA-ULM$^{LSGAN}$ to learn from a broader distribution of MB signals-specifically, MB signals that were not part of the LOCA-ULM$^{Experimental}$ training dataset, and improve the localization performance of the DECODE network.

## LOCA-ULM significantly improves the computational performance of MB localization

The computational performance of four localization algorithms (conventional ULM, Deep-ULM, mSPCN, and LOCA-ULM) was evaluated using 100 simulated imaging frames with different sizes of field-of-view (FOV). The FOV ranged from 0.40 $mm^2$ (16 pixel × 16 pixel area) to 139.88 $mm^2$ (300 pixel × 300 pixel area), with a 4-pixel increment step (in each dimension). The pixel resolution was fixed at 39.4 μm. For each FOV size, the number of MBs simulated in each frame also increased from 16 to 300 in increments of 4. Figure 2f presents the computational time results for the four different localization methods. Computational times encompass the MB localization process, which includes normalized cross-correlation and regional maximum search for conventional ULM, a network forward pass for LOCA-ULM, and a network forward pass with centroid calculation for both Deep-ULM and mSPCN. Conventional localization based on normalized cross-correlation exhibits a steep increase in computational time as the FOV expands (i.e., the number of pixels increases). In contrast, all deep learning-based methods demonstrate improved computational performance, with LOCA-ULM consistently showing the lowest computation time across all FOVs. At an FOV of 139.88 $mm^2$, the computational time for LOCA-ULM was approximately 1.7 s, representing a 5.3-fold acceleration compared to the conventional localization (9.1 s). Moreover, LOCA-ULM directly estimates the MB centers, therefore bypassing the process of peak identification required by Deep-ULM and mSPCN. This efficiency translates to LOCA-ULM achieving a twofold acceleration compared to mSPCN (3.6 s) and deep-ULM (3.5 s) for an FOV of 139.88 $mm^2$. All methods were evaluated on an NVIDIA RTX A6000 GPU.

## Considerations: evaluating the impact of conventional localization error on LOCA-ULM precision

LOCA-ULM demonstrated superior localization precision at high MB concentrations in our experiments. However, it is important to note that the simulation pipeline for LOCA-ULM used MB templates extracted from in vivo data, with peaks identified by the conventional localization method serving as the ground truth. Therefore, any MB localization errors generated from using conventional localization can propagate into the LOCA-ULM training dataset. To quantify the error propagation, we trained LOCA-ULM using Field II-simulated MB templates that included phase aberration errors (Supplementary Methods 1). Supplementary Fig. 1 displays the localization errors for both LOCA-ULM and conventional localization. The mean lateral and axial localization errors for LOCA-ULM ($\mu_x = 0.412$ μm, $\mu_z = -8.840 \mu m$) align closely with those for the conventional localization method ($\mu_x = 0.348 \mu m, \mu_z = -8.388 \mu m$). These findings suggest that the accuracy of LOCA-ULM in localizing isolated MBs is inherently limited by the ground truth localization estimate in the training phase. Thus, addressing PSF distortions (e.g., by phase aberration correction or other image-quality-enhancing beamforming methods) before localization remains an essential step to improving localization accuracy.

## LOCA-ULM demonstrates superior in vivo ULM imaging performance in a rat brain

We demonstrated the generalizability of LOCA-ULM using in vivo rat imaging datasets. Figure 3c, d shows the final ULM images based on 20,000 frames (a total of 80 s of data acquisition) of accumulation with MB injection rate of $15 \mu L$/min (Methods). As shown in the power Doppler image in Fig. 3a, the vascular bed in the rat brain presents large variations of vessel sizes, which indicates a broad distribution of MB flow rates and concentrations[8]. As shown in Fig. 3c, conventional localization suffered from poor localization performance in regions with high MB concentrations, which manifest as disconnected and missing vessels (red arrows in Fig. 3e). In contrast, LOCA-ULM revealed the dense cerebral vascular networks in these regions, which were well-perfused and fully connected (red arrows in Fig. 3f).

Next, we compared LOCA-ULM with the state-of-the-art MB localization method based on MB separation[10]. We used the MB separation filter to separate the ultrasound MB data into two subgroups: MBs flowing away from the transducer (downward flow) and flowing toward the transducer (upward flow), as shown in Fig. 3b. Figure 3g, h demonstrate that MB separation facilitated more robust MB localization and tracking in high MB density regions for both conventional and LOCA-ULM. The improvement is most significant for conventional localization, which suffered from poor MB localization performance in high-density MB regions without MB separation. The intersecting and adjacent small vessels that were missing by conventional localization now become clearly visible by using MB separation.

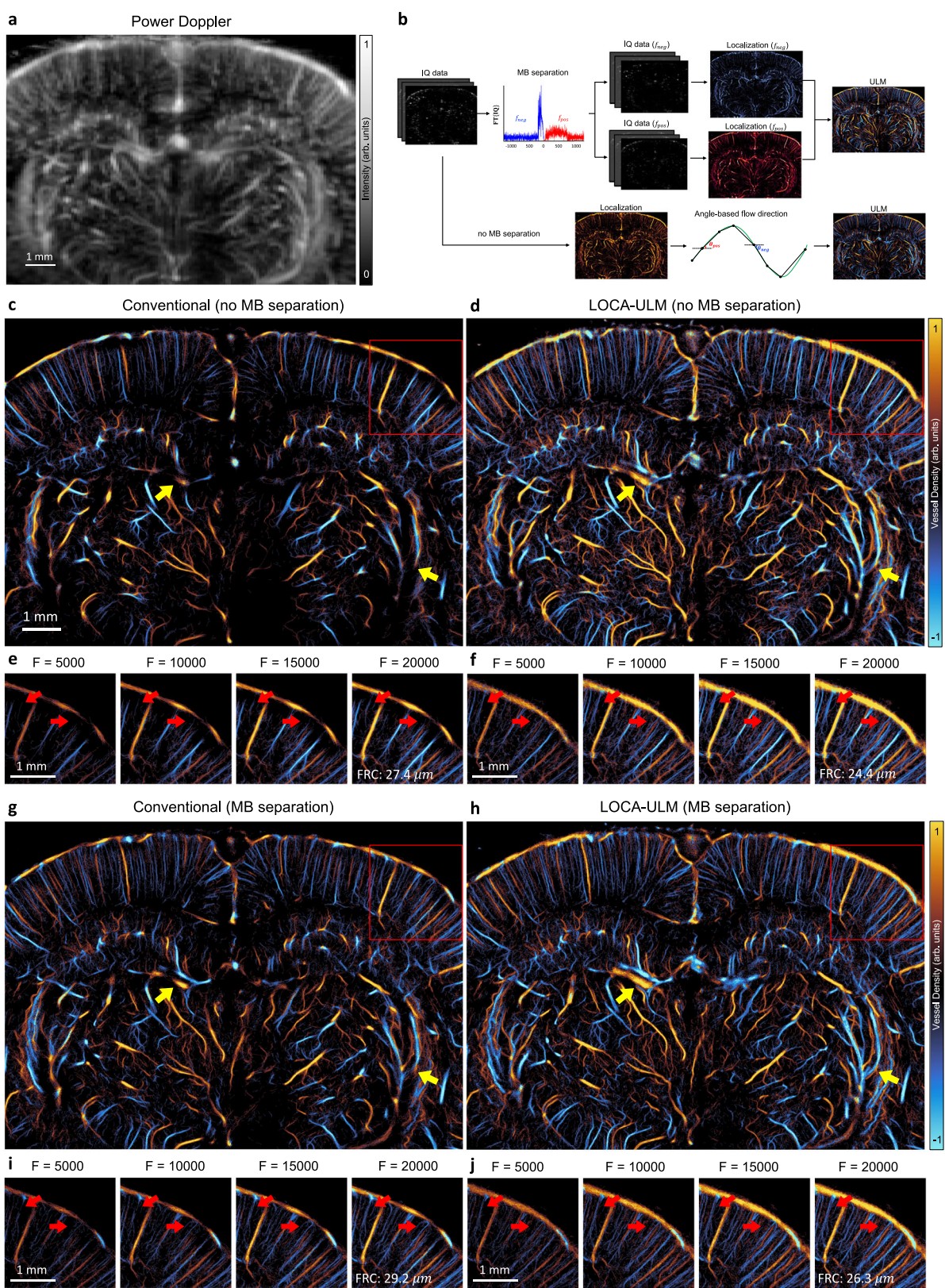

For LOCA-ULM, the improvement was moderate because LOCA-ULM was already efficient with localizing MBs in high-density regions. This is shown by comparing Fig. 3d, h, where most of the cerebral vasculature was consistent before and after applying MB separation for LOCA-ULM (indicated by yellow arrows). When comparing Fig. 3d, g, it becomes clear that even with MB separation, conventional localization still could not achieve a similar MB localization performance as LOCA-ULM

without MB separation. This is a notable finding because it suggests that LOCA-ULM alone can already outperform the state-of-the-art MB localization technique and does not require the assistance of post-processing methods such as MB separation.

Finally, the spatial resolution of the ULM reconstructions was measured by the Fourier Ring Correlation (FRC) method, which uses the track-splitting strategy and a 2-$\sigma$ threshold curve as proposed by

**Fig. 3 | Comparison of LOCA-ULM and conventional localization to in vivo rat brain ultrasound data. a** Power Doppler image generated by accumulating 2500 frames (a total of 10 s of acquisition, bregma: −4.4 mm) of rat brain ultrasound data. **b** In vivo rat brain localization workflow. The IQ data after tissue clutter filtering was processed with and without Fourier-based MB separation. For MB separation, the high-concentration MB dataset was divided into subsets of upward and downward flow towards the transducer using a directional filter[10]. Angle-based flow direction was used for the dataset without MB separation. For each dataset, MB locations was determined by performing normalized cross-correlation with an empirically

determined PSF function (i.e., conventional localization) or LOCA-ULM. The uTrack algorithm was used to pair the localized MB centers and estimate their trajectories. **c**–**j** Each ULM directional flow maps were generated by accumulating 20,000 frames (a total of 80 s of acquisition), **c, d** without MB separation and **g,h** with MB separation. **e, f, i, j** Improvement of vessel structures with respect to the increasing number of frames is displayed on the bottom, shown for the area marked with a red rectangle. *n* = 1 experiment. F indicates the number of frames used for ULM reconstruction, and FRC indicates Fourier ring correlation.

ref. 30. Our results showed that both LOCA-ULM and conventional localization produced a spatial resolution that is below a half wavelength at the imaging frequency of 15.625 MHz (that is, 49.28 $\mu m$) regardless of the application of MB separation (Supplementary Fig. 2).

## MB-specific characteristics improve the localization performance

An ablation study was conducted to evaluate the impact of MB-specific characteristics on MB localization performance within the LOCA-ULM simulation framework. Each MB characteristic used in this study, including MB brightness levels, MB movement and lifetime, and background ultrasound noise, was individually excluded from the framework to determine their impact on MB localization. As shown in Supplementary Fig. 3b, the absence of MB brightness variations resulted in a marked reduction in MB detection rate (Supplementary Fig. 3b green arrows), resulting in an incomplete ULM reconstruction. On the other hand, the exclusion of MB movement and lifetime mostly impacted the larger vessel regions with higher MB densities (Supplementary Fig. 3c, yellow and blue arrows). Finally, the absence of background noise led to an increase in false-positive MB localizations because a network trained without noise tends to misclassify background noise as MBs (Supplementary Fig. 3d, red arrows). These results underscore the importance of integrating MB characteristics into the training pipeline to introduce context-dependent features to the network, thereby mitigating the domain mismatch between training and testing datasets to facilitate high-fidelity ULM reconstructions.

## LOCA-ULM-based MB localization automatically adapts to different MB concentrations

In this study, we evaluated the performance of LOCA-ULM in vivo across different MB concentrations by increasing the MB injection rate from 20 $\mu L$/min to 40 $\mu L$/min (Methods). Figure 4 presents ULM images reconstructed for 20 $\mu L$/min and 40 $\mu L$/min injection rates using conventional ULM and LOCA-ULM in a rat brain, utilizing 25,000 frames (a total of 100 s of acquisition). Compared to conventional ULM, LOCA-ULM provided a superior reconstruction of the cerebral vasculature. At regions with high MB concentrations, such as large vessels, conventional ULM failed to reconstruct vessel structures due to a high MB missing rate (red arrows in Fig. 4a, c). LOCA-ULM, however, successfully revealed these large vessels missed by the conventional ULM (red arrows in Fig. 4b, d). Moreover, conventional ULM showed a decrease in the intensity of the reconstructed vessel with increased MB injection rates, leading to degraded vessel delineation. For example, two adjacent vessels separable at an MB injection rate of 20 $\mu L$/min (yellow arrows in Fig. 4a) became indistinguishable at an MB injection rate of 40 $\mu L$/min (yellow arrows in Fig. 4c). In contrast, LOCA-ULM maintained a clear separation of these vessels at both injection rates (yellow arrows in Fig. 4b, d). LOCA-ULM also delineated small vessels near the cortical surface that could not be clearly reconstructed by conventional ULM (green arrows in Fig. 4a–d). When comparing LOCA-ULM with the conventional ULM with state-of-the-art MB separation, LOCA-ULM provides improved separation of adjacent vessels with enhanced contrast in both 20 $\mu L$/min and 40 $\mu L$/min injection rates (green arrows in Fig. 4e–h). LOCA-ULM also improved the visualization of the vascular networks by more accurately

reconstructing the connected vessel structures branching from the main vessels (red arrow in Fig. 4e–h). Further comparison of LOCA-ULM with MB separation at 90 $\mu L$/min injection rate (Supplementary Fig. 4) shows that LOCA-ULM achieves robust reconstruction even in challenging conditions where conventional ULM with MB separation fails.

The efficacy of LOCA-ULM localization was further evaluated by a quantitative analysis that used the average Power Doppler (PD) intensity as the reference. As shown in Fig. 4i, the MB count of LOCA-ULM closely followed the trend of increasing PD intensity, while conventional localization did not. This indicates that conventional localization has already become saturated even at the lowest MB injection rate (20 $\mu L$/min). With the addition of MB separation (Fig. 4j), conventional localization showed improved localization efficacy. However, the number of MB localized plateaued at a low injection rate of 20 $\mu L$/min. In contrast, LOCA-ULM demonstrated superior localization performance, localizing approximately 1.5 times more MBs at low concentration (20 $\mu L$/min) compared to conventional localization, and showing a progressive increase in the number of localized MBs as the concentration increased. The quantitative results provide a good agreement with the ULM images, where LOCA-ULM reconstructed ULM images show increased microvessel intensity and contrast with increased MB injection rate (white dashed ROIs in Fig. 4f, h), while conventional ULM shows constant microvessel intensity despite the increased MB injection rate (white dashed ROIs in Fig. 4e, g).

In Fig. 5a, we compared the MB localization performance of LOCA-ULM with two other deep learning-based MB localization methods, Deep-ULM and mSPCN, on the high MB concentration rat brain dataset (40 $\mu L$/min) from the previous section. Both Deep-ULM and mSPCN reconstructed continuous vessel structures and revealed large vessels missed by conventional ULM (red arrows in Fig. 5a). However, Deep-ULM and mSPCN missed certain vessels that are clearly visible with LOCA-ULM (e.g., green arrows in Fig. 5a). Figure 5b shows the intensity profiles of vessels near the cortical surface (along the white line indicated in Fig. 5a), where LOCA-ULM successfully identified three vessels with distinct peaks, two of which were not discernible using other localization methods (marked by * in Fig. 5b). Moreover, with an equivalent number of imaging frames, Deep-ULM and mSPCN displayed less perfused vessels compared to LOCA-ULM, where LOCA-ULM demonstrated more than a twofold increase in vessel intensity (Fig. 5c), implying a higher MB detection rate.

## LOCA-ULM increases sensitivity for functional ultrasound localization microscopy (fULM)

Functional ULM (fULM) is a technique that combines ULM with functional ultrasound (fUS) to image brain-wide neurovascular activities on a microscopic scale[31,32]. fULM is a challenging technique because higher MB concentrations in the bloodstream are needed to provide higher sensitivity to hemodynamic responses, while MB concentration must also be kept low to facilitate robust MB localization and generate better ULM images. When a lower MB concentration is used, repeated stimulations and data acquisitions are typically needed to accumulate adequate MB signals and neural response measurements. This process elongates the data acquisition time and undermines the temporal resolution of fULM.

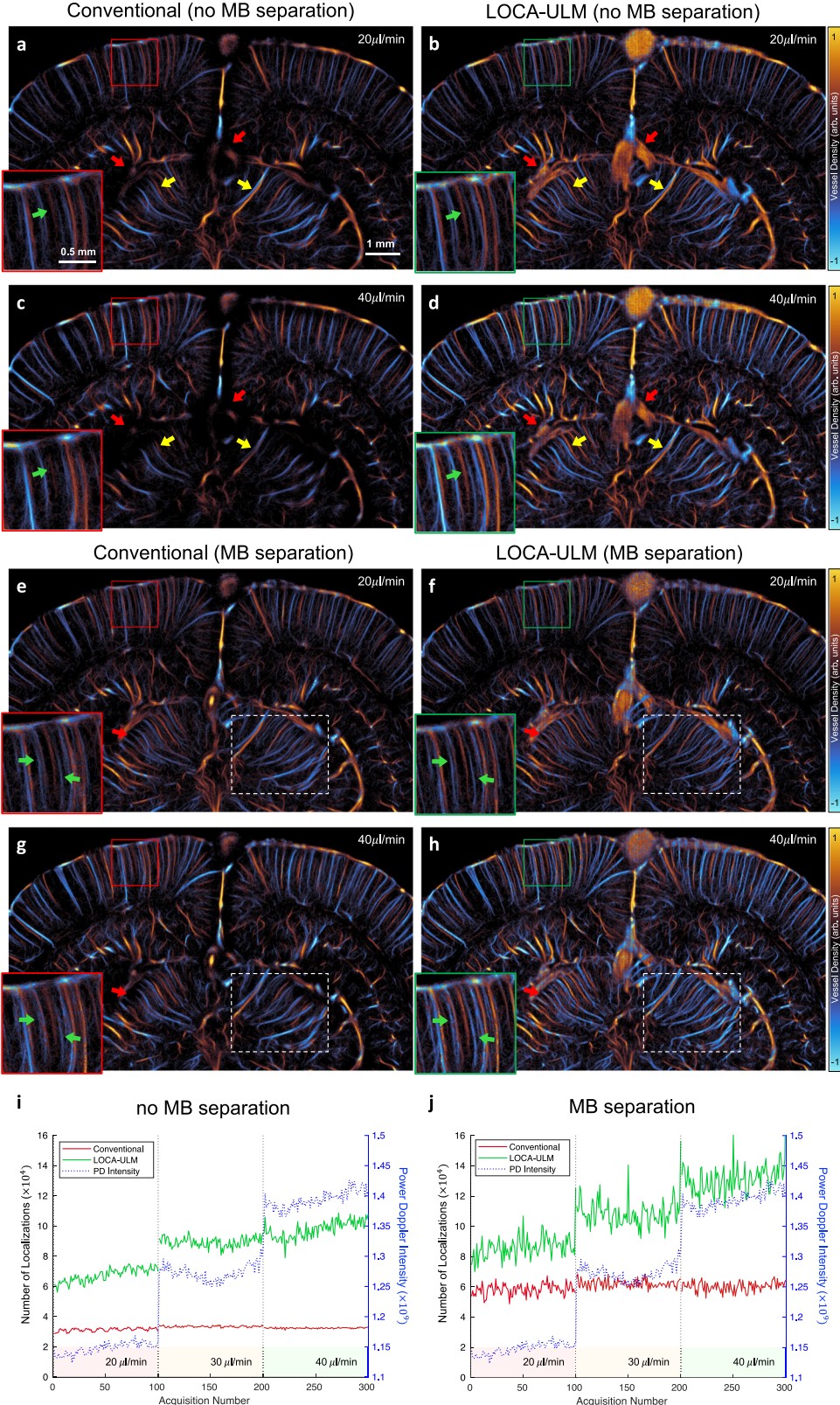

**Fig. 4 | Effect of different MB injection rates (20, 30, 40 µL/min) on LOCA-ULM and conventional localization for rat brain ULM imaging. a–h** Each ULM image was generated by accumulating 25,000 frames of ultrasound data (a total of 100 s of acquisition, bregma: −5.6 mm). **a–d** ULM reconstruction without MB separation and **e–h**, with MB separation. **i, j** Comparison of total MB count per acquisition (a total of 250 frames per acquisition) for LOCA-ULM and conventional localization at different MB injection rates ($20µL/$ min ,$30µL/$ min, and $40µL/$ min). Two data-sets, **i** without MB separation and **j** with MB separation. $n = 1$ experiment. Source data are provided as a Source Data file.

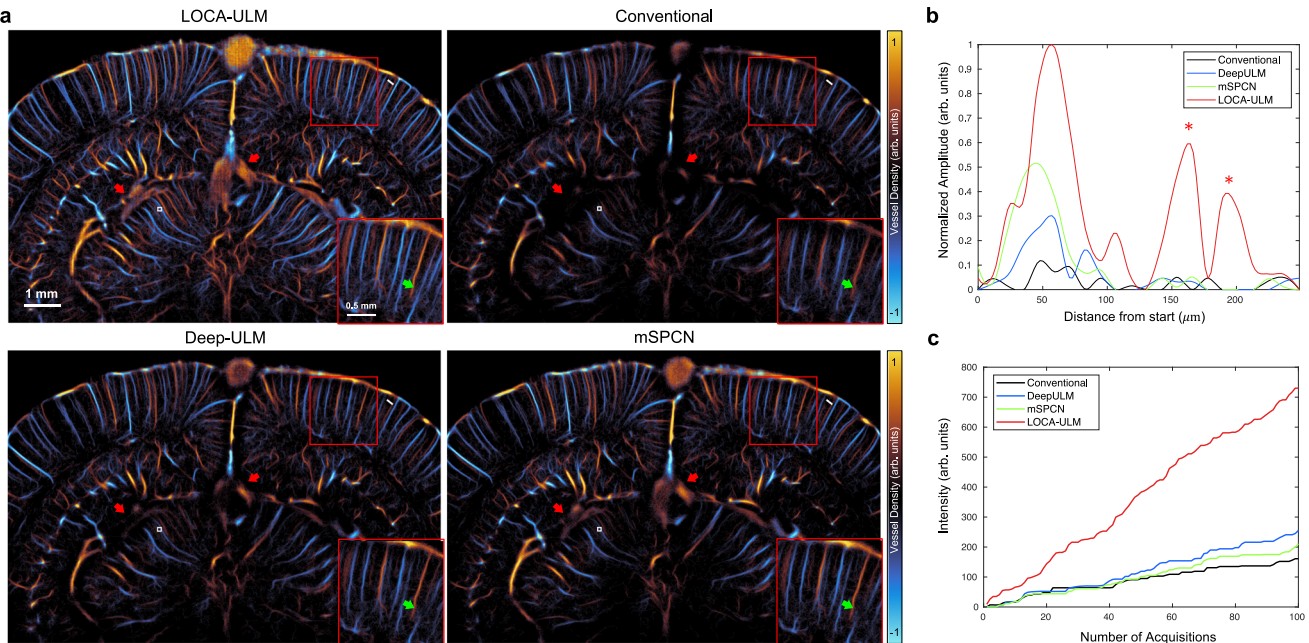

**Fig. 5 | Comparison among LOCA-ULM, conventional ULM, Deep-ULM[14], and mSPCN[21] in in vivo rat brain ultrasound data under high MB concentration (injection rate 40 μL/min). a** Each ULM image was generated by accumulating 25,000 frames of ultrasound data (a total of 100 s of acquisition, bregma: −5.6 mm) for MB injection rate of 40 μL / min. **b** Intensity profile of vessels indicated by a white line in (**a**). **c** Accumulated intensity of reconstructed vessel with respect to an increased number of frames indicated by the white box in (**a**). n = 1 experiment. Source data are provided as a Source Data file.

fULM provides an ideal testing scenario for LOCA-ULM because LOCA-ULM allows better ULM image reconstruction under high MB concentrations, which is essential for fULM. Here we used a similar experiment setup as ref. 31, where whisker stimulations on rats were used for fULM (details provided in Methods, Fig. 6a, b). For the fULM experiment, we used an MB injection rate of 60 μL/min to increase the MB concentration in the cerebrovasculature of the rat brain to increase fULM sensitivity. LOCA-ULM produced a superior ULM image under high MB concentration, showing enhanced contrast and finer vascular details compared to conventional ULM (white arrows in Supplementary Fig. 5). Figure 6d displays the MB count from pixel **i** and **ii**, which indicates significantly higher MB detection rate by LOCA-ULM. For pixel **i** in Fig. 6d, e, which was located within vessels close to the cortical surface, conventional localization produced sparse MB counts, ranging from 0 to 5 MBs/s. For smaller vessels that have a lower MB flow rate, such as pixel **ii** in Fig. 6d, e, the MB counts are also reduced, ranging from 0 to 3 MBs/s. On the other hand, LOCA-ULM produced a twofold increase in MB count, ranging from 0 to 11 per second for vessel **i** and 0 to 6 per second for vessel **ii** in Fig. 6d, e.

Figure 6e presents the fULM activation maps derived from both LOCA-ULM and conventional localization (detailed in Methods). LOCA-ULM presents strong activation in the vessels within the barrel field of the primary somatosensory cortex (S1BF) and the ventral posterior medial nucleus (VPM)−regions corresponding to whisker stimulation. As illustrated in Fig. 6c, functional ultrasound (fUS) demonstrates enhanced sensitivity in identifying the increased blood flow, detecting more extensive areas of activation compared to fULM using conventional localization. In contrast, the activated regions detected by fULM with LOCA-ULM corresponds closely with the areas identified by the fUS, while also offering a significant improvement in spatial resolution. When compared to conventional MB localization, LOCA-ULM demonstrated a 1.85-fold increase in activated pixel count in the S1BF and VPM regions (red ROIs in Fig. 6e). As shown by the blue arrows in Fig. 6e, conventional MB localization was unable to reconstruct the complete vasculature under high MB concentrations and failed to display vascular responses that are clearly visualized in

LOCA-ULM. Furthermore, LOCA-ULM revealed strong activation in smaller vessels (approximately 30-μm diameter), as indicated by red arrows in Fig. 6e. In contrast, these vessels did not show strong responses using conventional MB localization, likely due to missing MB localizations.

Figure 6f illustrates the evolution of activation maps with an increasing number of stimulation cycles, which is a typical method of increasing fULM sensitivity to hemodynamic responses[31]. It can be observed that the combination of high-concentration MBs and robust MB localization provided by LOCA-ULM can reduce the number of stimulation cycles required for fULM. For example, Fig. 6f shows that the activation maps produced by LOCA-ULM after six cycles of repetition exhibit strong and consistent responses in the S1BF and VPM regions, which are comparable to the response detected by accumulating 15 cycles of repetition by LOCA-ULM. In contrast, a high number of stimulation cycles was necessary for conventional localization-based fULM because of the lower MB detection efficacy that results in reduced sensitivity to neural responses. When comparing activation maps from conventional ULM and LOCA-ULM with an increasing number of repetitions, LOCA-ULM consistently outperformed conventional ULM (Fig. 6f). Notably, even with 15 cycles of repetition, conventional ULM still could not produce similar quality activation maps to LOCA-ULM with only six cycles. These results demonstrate that the enhanced MB localization performance provided by LOCA-ULM is beneficial for improving fULM sensitivity to hemodynamic responses evoked by neural activities.

## Discussion

In this study, we presented a context-aware deep learning-based MB localization method (LOCA-ULM) along with an LSGAN-based MB simulation pipeline to facilitate high-quality ULM imaging under high MB concentrations. We designed a contrast-enhanced ultrasound simulation workflow to produce in vivo-like simulation data with ground truth for training, accounting for various factors, including MB shape and brightness, background ultrasound noise, and temporal movement of MBs across multiple imaging frames. Our study adopted

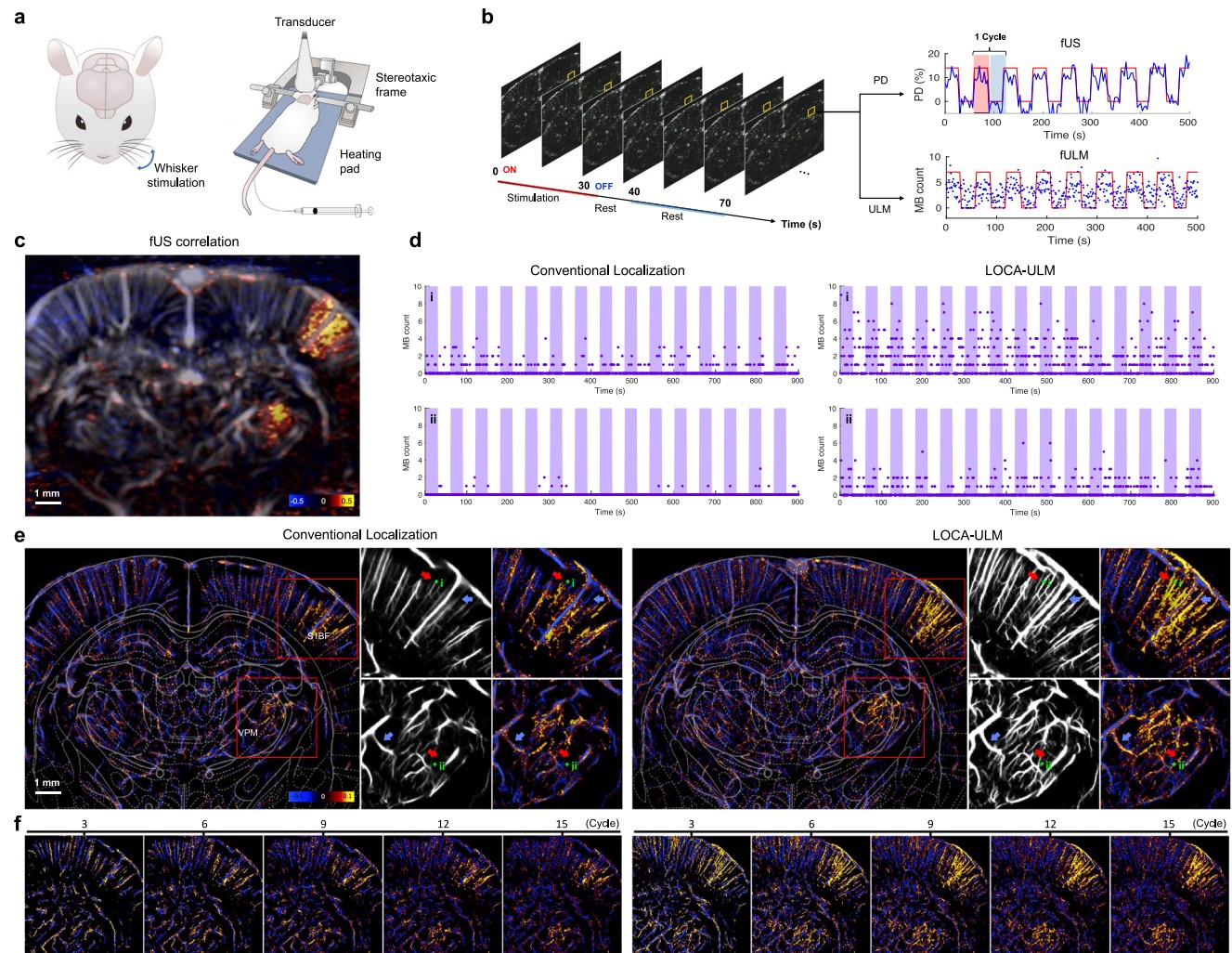

**Fig. 6 | LOCA-ULM increases MB signal sensitivity to blood flow during brain activation. a**, **b** Schematic of the experimental setup for functional ultrasound (fUS) and functional ULM (fULM) brain imaging conducted in the coronal plane (bregma: −3.3 mm). The fUS and fULM experiment involved whisker stimulation in an anesthetized rat with continuous intravenous injection of MBs (Methods). **c** fUS activation map calculated as the Pearson correlation coefficient between Power Doppler signals over time and the stimulation pattern. **d** MB count over time for a pixel in selected vessels (i and ii in **e**) comparing conventional ULM and LOCA-ULM. **e** fULM activation map calculated as the Pearson correlation coefficient between the MB count (after tracking) over time and the stimulation pattern for both conventional ULM and LOCA-ULM. Zoomed-in views of the ULM image and activation map in the barrel cortex (S1BF) and the ventral posteromedial nucleus (VPM) are shown, marked by a red rectangle. **f** Progressive enhancement of the fULM activation maps with an increasing number of stimulation cycle repetitions. $n = 1$ experiment. Source data are provided as a Source Data file.

the concept of DECODE designed for SMLM as the MB localization network for ULM.

The LOCA-ULM simulation pipeline effectively mitigates the domain discrepancy commonly encountered with supervised deep learning-based MB localization techniques by using LSGAN-generated MB templates. In practice, the inherent complexity of modeling ultrasound MB PSFs in vivo makes it challenging to construct theoretical PSF models that resemble real MB signals. When trained with Gaussian MB templates, all the deep learning-based MB localization methods (Deep-ULM, mSPCN, and LOCA-ULM) exhibited suboptimal localization performance, which manifested as the presence of gridding artifacts and poor vessel reconstructions (Supplementary Fig. 6). However, when LOCA-ULM was trained with MB templates generated by LSGAN, it demonstrated enhanced ULM reconstruction over those trained by Gaussian and Field II MB templates. This improvement is characterized by enhanced contrast, reduced gridding artifacts, and more comprehensive reconstruction of vascular structures for the chicken embryo chorioallantoic membrane (CAM) and rat brain (Supplementary Fig. 7). Moreover, LOCA-ULM trained with

LSGAN-generated MBs presented higher vessel filling (VF) percentage and faster vessel saturation rate over LOCA-ULM<sup>Experimental</sup> and conventional localization (Fig. 2e). However, it is important to consider the spatial variations of MB signals and their potential impact on localization accuracy. As illustrated in Supplementary Fig. 8c, our brain-tissue mimicking phantom experiment indicates that MB signals do present depth-dependent variations (Supplementary Methods 4). However, the MB templates generated by LSGANs follow a similar distribution as experimental MB signals across the entire FOV (Supplementary Fig. 8d). Therefore, the training data using LSGAN-generated MBs does include a complete MB distribution from different imaging depths, facilitating a robust MB localization performance for LOCA-ULM. It should be noted that the performance of other transducer arrangements (e.g., curvilinear) was not investigated in this study.

LOCA-ULM further enhances MB localization performance by integrating MB-specific characteristics into the simulation framework (Supplementary Fig. 3). Trained with our comprehensive simulation pipeline that mimics the ultrasound imaging process, LOCA-ULM

improved the localization efficacy and reduced MB missing rate across all tested concentrations for both simulation and in vivo data. As a result, LOCA-ULM provided clear delineation of densely populated large vessels and adjacent microvessels that were missed by conventional ULM and reconstructed complex vascular networks with connected structures (Figs. 3, 4). Notably, LOCA-ULM maintained robust localization efficiency even at high concentrations (40 μL/min of MB injection rate), achieving up to a threefold increase in localized MBs over conventional localization without compromising spatial resolution (Fig. 4i).

In this study, LOCA-ULM was benchmarked against current deep-learning MB localization methods, demonstrating improved localization efficiency. Unlike Deep-ULM and mSPCN, which apply loss functions based on least-squares regression under the $l_1$ regularization[14,21], LOCA-ULM uses a joint loss function optimizing both MB count loss and localization loss to achieve MB localization in an end-to-end manner. The count loss encourages the network to output a sparse and high-probability detection map, providing complementary information about the position of each MBs. In turn, the localization loss directly predicts the MB locations by fitting a sum of Gaussian distributions weighted by the detection probability. It was shown in the simulation study that LOCA-ULM reduced the average missing rate by twofold when benchmarked against Deep-ULM and mSPCN (Fig. 2b). The improvement was consistent in in vivo imaging where LOCA-ULM reconstructed well-perfused vascular networks compared to other deep learning-based techniques (Fig. 5a, b) and achieved more than twofold enhancement in vessel intensity (Fig. 5c).

We have also demonstrated the effectiveness of LOCA-ULM in achieving both high-speed processing (Fig. 2f) and accelerated data acquisition for ULM. In theory, higher MB concentration facilitates faster ULM imaging by accelerating the MB filling rate of smaller vessels, which translates to shorter data acquisition time[8]. Our experiment across various in vivo datasets has shown LOCA-ULM maintains robust performance even at extreme MB concentrations, enhancing the achievable localization concentration (Fig. 4i, j and Supplementary Fig. 4). This robust performance reduces the need for additional techniques that increase the computational load, such as MB separation, typically required by conventional methods (Figs. 3, 4).

The enhanced localization performance at high MB concentrations makes LOCA-ULM a promising tool for fULM, a technique that demands high efficacy in localizing MBs within dense distributions. LOCA-ULM increased the sampling of the temporal MB count signals, resulting in increased fULM sensitivity to neural responses (Fig. 6d) and a higher level of activation detected from smaller vessels with slower blood flow (Fig. 6e). Importantly, the increased fULM sensitivity from LOCA-ULM reduces the number of stimulation cycles for fULM (Fig. 6f), crucial for the practical applications and especially in studies involving moving animals where repeated stimulations are challenging. Furthermore, LOCA-ULM could potentially improve dynamic ULM (DULM)[33] by increasing the number of localized and tracked MBs, offering more velocity measurements over time, and potentially decreasing the total acquisition time by relaxing the need for data acquisitions from multiple cardiac cycles.

The concept of using LSGANs to generate realistic MB signals can be extended to a broad range of ultrasound imaging applications associated with MBs. Our proposed simulation pipeline does not require any prior knowledge of the MB PSF model or ultrasound imaging settings to create the training dataset. Our method can be easily used to create simulated data, which can aid in robust training and reduce the challenge of generalizing deep learning-based localization to in vivo ultrasound data. However, the DECODE network and LSGAN do need to be retrained when the ultrasound imaging settings are altered. In addition, a stable training of LSGAN requires a large collection of spatially isolated MB signals extracted from experimental data, which can be pragmatically challenging. Furthermore, errors in

single MB localization from conventional methods could potentially propagate during the training of the LOCA-ULM network (Supplementary Fig. 1). Also, the performance of LOCA-ULM may be undermined by inaccurate simulation parameters (e.g., MB brightness levels, background noise, etc.), resulting in suboptimal MB localization performance. Nevertheless, because LOCA-ULM outputs uncertainties of localizations, one can use the predicted uncertainties to reject unreliable localizations.

## Methods
### Simulation pipeline
The simulated datasets for training are generated during the network training, creating 10,000 frames per epoch, and using each frame only once for training. Because LOCA-ULM is trained purely on simulated data, it may fail to generalize to real ultrasound data if there is a discrepancy between the two datasets. To address this issue, we created a realistic model for the ultrasound image formation process that incorporates LSGAN-generated MB signals and data-informed ultrasound background noise (Fig. 1a). Compared with the generic GAN, LSGAN replaces the sigmoid cross entropy loss in the discriminator with a least-squares loss, facilitating the generator to create more realistic images and learn the distribution of the training data more robustly[26]. LSGAN has been applied in medical imaging to improve spatial resolution and prevent mode collapse (i.e., generator creating limited ranges of outputs)[34–36]. The training problem for the LSGAN can be formulated as:

$$\min_{D} \mathcal{L}(D) = \frac{1}{2} \mathbb{E}_{\mathbf{x} \sim p_{data}(\mathbf{x})} \left[ (D(\mathbf{x}) - 1)^2 \right] + \frac{1}{2} \mathbb{E}_{\mathbf{z} \sim p_z(\mathbf{z})} \left[ (D(G(\mathbf{z}))^2 \right]$$
$$\min_{G} \mathcal{L}(G) = \frac{1}{2} \mathbb{E}_{\mathbf{z} \sim p_z(\mathbf{z})} \left[ (D(G(\mathbf{z})) - 1)^2 \right] \tag{1}$$

where $D$ denotes the discriminator, $G$ represents the generator, $\mathbf{z}$ represents the input signal, which was randomly sampled from a normal distribution, and $\mathbf{x}$ represents the MB templates extracted from real ultrasound images.

To collect the LSGAN training data, the in vivo ultrasound images were first interpolated by a factor of 5 (5×) in the axial dimension and 10× in the lateral dimension. This corresponds to a 0.064 λ pixel size for the CAM images and 0.1 λ pixel size for rat brain images (Supplementary Table I, MB template pixel resolution). Square patches (65pixel × 65pixel) were extracted from the in vivo ultrasound images and used to create the simulated images for training. Each patch contains a single MB signal that takes the peak location identified by the normalized cross-correlation (NCC) localization algorithm as the true MB location[27]. A total of 3000 patches were manually selected from the in vivo ultrasound images to train the LSGAN and the mean ($\mu_{I_{max}}$) and standard deviation ($\sigma_{I_{max}}$) of the maximum intensity were calculated. After training, the synthetic MB templates generated from the LSGAN were saved into a bank of MBs. To generate training data for the network, a list of ground truth MB positions was sampled in sub-wavelength pixel resolution (Supplementary Table I, DECODE output pixel resolution) and convolved with randomly selected MB template retrieved from the bank of MBs.

In our simulation pipeline, we modeled MB flow within vessels using a motion model that accounts for random changes in MB movement. The initial positions of MBs were randomly sampled from a uniform distribution within the dimensions of the imaging FOV. The lifetime of each MB ranged from 1 to 20 frames, randomly sampled from a uniform distribution. Each MB is assigned an initial speed and a two-dimensional (2D) unit vector that represents the initial direction. The speed was randomly sampled from a uniform distribution between 5 mm/s and 25 mm/s. To simulate stochastic MB motion, the direction of the MBs was perturbed at every time step by adding a small random vector. The components of this random vector were

sampled from a normal distribution with a standard deviation of 0.2. After the random direction perturbation, the direction vector and the speed are multiplied to obtain the velocity. The MB position was updated according to the velocity, and the position at each time frame was stored as the ground truth. The brightness level of the MB was determined from a Gaussian distribution $N(\mu_{I_{max}}, \sigma_{I_{max}})$ and remained constant for the duration of the MB's lifetime. 80 pixel × 80 pixel sized simulated frames were created and the images are down-sampled by a factor of 2 to create the final 40 pixel × 40 pixel sized training dataset (Fig. 2b, LSGAN).

### Background noise modeling

To add realistic electronic noise to the simulation, we used Rician distribution as the noise model in this study. Assuming an additive Gaussian noise in both real and imaginary parts of the in-phase quadrature (IQ) data, the B-mode signal $I_{x,z}$ (i.e., magnitude of IQ at pixel $(x,z)$) satisfies the distribution:

$$P(I_{x,z}|\nu_{x,z},\hat{\sigma}_{x,z}) = \frac{I_{x,z}}{\hat{\sigma}_{x,z}}\exp\left(\frac{-\left(I_{x,z}^2+\nu_{x,z}^2\right)}{2\hat{\sigma}_{x,z}^2}\right)I_o\left(\frac{I_{x,z}\nu_{x,z}}{\hat{\sigma}_{x,z}^2}\right), \quad (2)$$

where $\nu_{x,z}$ is the magnitude of the B-mode signal at pixel $(x,z)$ without noise, $\hat{\sigma}_{x,z}$ is the standard deviation of the additive noise, and $I_o$ is the modified Bessel function of the first kind with order zero. In this study, the $\hat{\sigma}_{x,z}$ was estimated experimentally by taking the temporal mean of the acquired electronic noise data $E(x,z,t)$ as,

$$\hat{\sigma}_{xz} = \sqrt{\frac{2}{\pi}}\frac{1}{N}\sum_{t=1}^{N}E(x,z,t), \quad (3)$$

where $N$ is the number of samples considered for estimation. Electronic noise in ultrasound images were obtained by performing the same ultrasound acquisition as the in vivo experiment without any imaging target (e.g., in air) (Fig. 1c).

### DECODE architecture

Accurate and robust MB localization under a wide range of vessel sizes and MB concentrations is essential for successful ULM. Inspired by the previous study by ref. 24, we implemented the DECODE network that enables simultaneous detection and localization of MBs in a probabilistic framework. Several key aspects allow DECODE to outperform conventional localization methods. First, DECODE can improve detection and localization accuracy by capturing the temporal context of the MB flow. The architecture is divided into two networks: a frame analysis network that comprises three separate U-Nets, where features of three consecutive frames are extracted in each U-net. The frame analysis network is followed by a *temporal context network*, where the final outputs of the three U-Nets are combined to capture the temporal context information between neighboring frames (Fig. 1e).

Moreover, the DECODE network was trained to minimize the total loss that consists of three parts: an MB count loss ($\mathcal{L}_{count}$), MB localization loss ($\mathcal{L}_{loc}$) and a background loss ($\mathcal{L}_{bg}$)[24]. The MB count loss is represented by a Bernoulli distribution $p_k$ that indicates the probability of detecting a microbubble near pixel $k$. Given that the probability $p_k$ varies among the pixels, the mean ($\mu_{count}$) and variance ($\sigma_{count}^2$) of the Poisson-binomial distribution is given as $\mu_{count} = \sum_{k=1}^{K}p_k, \sigma_{count}^2 = \sum_{k=1}^{K}p_k(1-p_k)$, where $K$ is the total number of pixels. When $K$ is sufficiently large, the Poisson-binomial distribution approximates the Gaussian distribution defined as,

$$P(E|\mu_{count},\sigma_{count}^2) = \frac{1}{\sqrt{2\pi}\sigma_{count}}\exp\left(-\frac{1}{2}\frac{(E-\mu_{count})^2}{\sigma_{count}^2}\right) \quad (4)$$

where $E$ is the true number of simulated MBs. The log probability of $E$ is maximized when the $\mu_{count}$ approximates to $E$, equivalent to minimizing,

$$\mathcal{L}_{count} = \frac{1}{2}\frac{(E-\mu_{count})^2}{\sigma_{count}^2} + \log\left(\sqrt{2\pi}\sigma_{count}\right) \quad (5)$$

The localization loss is designed jointly to optimize the output variables of the Gaussian mixture model (GMM) to approximate the true posterior with respect to MB locations and brightness. A Gaussian distribution for each pixel $k$, weighted by the detection probability is used to approximate the true posterior. The four-dimensional Gaussian $P(\mathbf{u}_k|\boldsymbol{\mu}_k,\Sigma_k)$ is modeled as a distribution over the coordinates and brightness of the MB $\mathbf{u} = [x,y,z,I]$:

$$P(\mathbf{u}|\boldsymbol{\mu}_{kk},\Sigma_k) = \frac{1}{\sqrt{(2\pi)^4 \det(\Sigma_k)}}\exp\left(-\frac{1}{2}(\boldsymbol{\mu}_k-\mathbf{u})^T\Sigma_k^{-1}(\boldsymbol{\mu}_k-\mathbf{u})\right), \quad (6)$$

where $\boldsymbol{\mu}_k = [x_k+\Delta x_k, y_k+\Delta y_k, z_k+\Delta z_k, I_k]$ and $\Sigma_k = \text{diag}(\sigma_{x,k}^2, \sigma_{y,k}^2, \sigma_{z,k}^2, \sigma_{I,k}^2)$. The $(x_k,y_k,z_k)$ coordinate represents the center of pixel $k$, and $(\Delta x_k, \Delta y_k, \Delta z_k)$ is the sub-wavelength coordinates of the MB with respect to the center of pixel $k$. The distance between the inferred posterior and the true posterior is minimized (i.e., by minimizing the forward KL divergence) by optimizing the log-likelihood of the weighted Gaussian distributions over the ground truth (GT) MBs,

$$\mathcal{L}_{loc} = -\frac{1}{E}\sum_{e=1}^{E}\log\sum_{k=1}^{K}\left(\frac{p_k}{\sum_j p_j}\right)P(\mathbf{u}_e^{GT}|\boldsymbol{\mu}_k,\Sigma_k), \quad (7)$$

where $e$ represents each ground truth MB present in the image. The localization loss maximizes the likelihood of the ground truth positions and brightness $\mathbf{u}_e^{GT}$ over all predicted detections. The DECODE network was designed to output the nine parameters of the weighted Gaussian distribution with respect to the center frame of the three consecutive imaging frames: (1) probability $p_k$ that a MB was detected near pixel $k$, (2) the relative coordinates of the localized center $\Delta x_k, \Delta y_k, \Delta z_k$ respect to the pixel center $(x_k,y_k,z_k)$, (3) estimated brightness of the MB ($I$), (4) the uncertainties $\sigma_{x,k}, \sigma_{y,k}, \sigma_{z,k}, \sigma_{I,k}$, and (5) the background intensity ($B$). In this study, we used a 2D variant of DECODE to process the 2D ultrasound data. Also, the background loss ($\mathcal{L}_{bg}$) in DECODE was set to 0 since the background in ultrasound images was modeled separately using the noise model.

The DECODE network In Fig. 1e reveals the detailed architecture, where the U-Nets in the frame analysis and temporal context networks consist of two downsampling and upsampling layers. The convolution blocks in both networks adopted a kernel of 3 × 3 size followed by an Exponential Linear Unit (ELU) as an activation layer. The number of filters increases from 48, 96, and 192 for each downsampling layer, with the feature map size halved. The number of filters decreases from 192, 96, and 48 for each upsampling layer, with the feature map size doubled. The input of the DECODE network were ultrasound images upsampled to 2.5 × in axial dimension and 5 × in lateral dimension (Supplementary Table I, DECODE network input pixel resolution). At the inference stage, to enhance the precision of LOCA-ULM, localizations with the highest inferred uncertainties were eliminated.

### Evaluation metrics

We compared three evaluation metrics to measure the MB localization performance of LOCA-ULM and conventional localization in the simulation study. MB detection accuracy measures the fraction of correct localizations (within 5 pixels or $0.32\lambda$ the ground truth

position) among all localized MBs:

$$\text{MB Detection Accuracy} = \frac{\text{TP}}{\text{TP}+\text{FP}}, \qquad (8)$$

where TP is true positives and FP is false positives. The MB miss rate measures the fraction of missed localizations among all ground truth positions:

$$\text{MB Miss Rate} = \frac{\text{FN}}{\text{TP}+\text{FN}}, \qquad (9)$$

where FN is false negative. The localization error (L) computes the averaged root mean-squared distance between the correctly localized MBs (i.e., TP) and the corresponding ground truth MB positions.

$$L = \sqrt{\frac{1}{\text{TP}} \sum_{i \in \text{TP}}^{\text{TP}} \frac{(\hat{x}_i - x_i)^2 + (\hat{y}_i - y_i)^2}{2}}, \qquad (10)$$

where $x_i, y_i$ are the ground truth coordinates and $\hat{x}_i, \hat{y}_i$ are the predicted coordinates.

For quantitative assessment of the localization performance in in vivo CAM imaging, we calculated the vessel filling (VF) percentage using the method described by ref. 37. First, a region of interest (ROI) was carefully selected to include matching vascular structures in both imaging modalities. For optical microscopy images, vascular structures were manually segmented and used as the reference. This manual segmentation was executed using the GNU Image Manipulation Program (GIMP). The resulting optical binary map was then spatially registered with the reconstructed ULM image, which involved additional resizing and cropping to achieve proper alignment. The vessel filling (VF) percentage was calculated as,

$$\text{VF(\%)} = \frac{N_{\text{GT} \cap \text{ULM}}}{N_{\text{GT}}} \times 100, \qquad (11)$$

where $N_{\text{GT}}$ is the total number of pixels classified as the ground truth vessels in the optical image. $N_{\text{GT} \cap \text{ULM}}$ is the total number of pixels correctly classified by ULM with respect to the ground truth $N_{\text{GT}}$.

## ULM implementations

For each ULM dataset, an SVD-based clutter filter was applied to extract the MB signal from the surrounding tissue[38,39]. To reduce the intensity variations of the MB signal, all frames were normalized to a scale of 0 to 1 with respect to the minimum and maximum intensity within each acquisition (1600 frames for the CAM study, 250 frames for the rat brain study, 800 frames for the fULM study). Also, due to the hyperechogenicity of MBs, thresholding between the values of $0.1 - 0.2$ was selected empirically to remove low-intensity background and noise. After image processing, the images were upsampled to avoid the quantization artifacts associated with DECODE localization[24]. Then, the network was trained to output super-resolved locations with sub-wavelength resolution (Supplementary Table I, DECODE network output pixel resolution). For conventional ULM, NCC-based MB localization was employed using a pre-defined bivariate Gaussian distribution[27]. The centroid coordinates obtained by different localization methods were input into the uTrack algorithm[40]. uTrack solves tracking as a two-step linear assignment problem. The first step involves establishing frame-to-frame MB linking to generate initial track segments. In the second step, uTrack connects these track segments across the entire time-lapse sequence to form complete trajectories. The MB tracks were generated with a minimum persistence of ten frames. Furthermore, links between track segments with a linking angle exceeding 45° were considered unreliable and rejected.

## In vivo ULM data acquisition

All animal experiments were approved by the Institutional Animal Care and Use Committee (IACUC) at the University of Illinois Urbana-Champaign (IACUC Protocol number 22165). A total of four eight to twelve-week-old female Sprague Dawley rats (Charles River Laboratories, Inc.) were used in the rat brain and functional ULM study. The Chicken Embryo Chorioallantoic Membrane (CAM) study adhered to our institute's guidelines and was consistent with the NIH PHS policy on avian embryos and live vertebrate animals.

**Chicken embryo chorioallantoic membrane (CAM) study.** For our study, we used the CAM microvessel model and optical imaging to provide the reference ground truth for validating different MB localization methods. The ex ovo CAM microvessels present a dense and diverse network of blood vessels with diameters ranging from 10 to 155 $\mu m$ [41]. The optical transparency of CAM allows for robust registration between ultrasound and optical images and is therefore widely used as a benchmark for assessing microvasculature with contrast-enhanced ultrasound[10,42–44]. In this study, fertile chicken eggs were obtained by the University of Illinois Poultry Research Farm and kept in tilting incubators (Digital Sportsman Cabinet Incubator 1502, GQF Manufacturing Inc., Savannah, Georgia). After four days, the eggshells were removed, and the CAM embryos were mounted into a plastic holder in a position suitable for imaging. Then, the embryos were incubated for an additional 13 days in a humidified incubator (Darwin Chambers HH09-DA) until the desired developmental stage. A borosilicate glass tube (B120-69-10, Sutter Instruments, Novato, CA, USA) was pulled at high temperature and cut using a PC-100 glass puller (Narishige, Setagaya, Japan) to create a fine glass capillary needle for MB injection. 50 µL boluses of Definity® solution (Lantheus, Bedford, MA) were injected into the surface bloodstream of the CAM via the glass needle.

CAM ultrasound imaging was performed using the Vantage 256 system (Verasonics Inc., Kirkland, WA, USA) system with a high-frequency linear array transducer (L35-16vX, Verasonics Inc., Kirkland, WA). The transducer was placed at the side of the plastic holder to image the CAM surface through a lateral acoustic window. Optical microscopy was conducted simultaneously using a Nikon SMZ800 stereomicroscope (Nikon, Tokyo, Japan) with a DS-Fi3 digital microscope camera (5.9-Mpixel CMOS image sensor, Nikon). The optical microscope was positioned above the weigh-boat and aligned to register the optical image with the ultrasound imaging plane. The initial positioning of the ultrasound-optical plane was determined using real-time B-mode imaging of native red blood cell scattering at a high transmit voltage (30 V). Following this, the voltage was reduced to 6 V, and the chicken embryo was injected with microbubbles in preparation for contrast-enhanced imaging. Ultrasound data were obtained by using a 9-angle compounding plane-wave imaging sequence (step size of 1°) with a center frequency of 20 MHz, pulse repetition frequency (PRF) of 40 kHz, and a post-compounding frame rate of 1000 Hz. IQ data of 1600 frames per acquisition with a total of 20 acquisitions were generated (total 32 s of acquisition).

**Rat brain study.** Animals were anesthetized with isoflurane (5% induction, 1.5% maintenance) throughout the experiment. Before craniotomy, the jugular vein was catheterized, and then the animal was fixed on a stereotaxic frame. The scalp was removed, and the skull was thinned using a rotary micromotor with a 0.5 mm drill bit (Foredom K.1070, Bethel, CT). The skull was removed with the size of the cranial window of 12 mm (left-right) by 6 mm (rostral-caudal) below the bregma. To image the rat brain, Definity® MBs were diluted with saline to yield an initial concentration of $1.44 \times 10^9$ bubbles per ml. The diluted MBs were continuously infused using a syringe pump (NE-300, New Era Pump Systems Inc., Farmingdale, NY).

In the rat brain study (illustrated in Fig. 3), we used an injection rate of 15 $\mu L$/min. For the study of comparing the performance of different localization methods in different MB concentrations (illustrated in Figs. 4, 5), we varied the injection rate to 20, 30, and 40 $\mu L$/min with a 3-min waiting period after changing the injection rate to stabilize the systemic MB concentration. All rat brain data were acquired using a high-frequency linear array transducer (L22-14vX Verasonics Inc., Kirkland, WA) connected to a Vantage 256 system. Ultrasound data were obtained by using a 5-angle compounding plane-wave imaging sequence (step size of 1°) with a center frequency of 15.625 MHz, PRF of 28.57 kHz, and post-compounding frame rate of 1000 Hz. IQ data of 250 frames per acquisition with a total of 100 acquisitions were generated (total 100 s of acquisition). In the high-concentration rat brain study (illustrated in Supplementary Fig. 4), we used an injection rate of 90 $\mu L$/min. IQ data of 800 frames per acquisition were acquired per second.

**Functional ultrasound localization microscopy (fULM).** For the fULM imaging study, ketamine (Zoetis) and xylazine (AnaSed) anesthesia was employed during the imaging session to mitigate the vasodilation effect induced by isoflurane and the decrease of neurovascular coupling associated with isoflurane anesthesia[45–47]. Following the craniotomy, the target imaging plane, including both S1BF and VPM at the bregma coordinate of −3.3 mm, was identified using 2D Power Doppler ultrasound imaging. After identifying the imaging plane, the rats were administered a mixed solution of ketamine (40 mg/kg body weight) and xylazine (3 mg/kg body weight) in saline through intraperitoneal injection. Then, fUS data acquisition was performed using an L22-14vX high-frequency transducer connected to a Vantage 256 ultrasound system. fUS data were acquired using a 9-angle compounding plane-wave imaging sequence (step size of 1°) with a 40 V transmit voltage. The transmit frequency was set to 15.625 MHz with a transmitting PRF of 28.57 kHz and a post-compounding frame rate of 1000 Hz. 250 frames of post-compounding IQ data were acquired per second, and a total of 180 acquisitions were acquired (a total of 3 min of acquisition). SVD-based spatiotemporal clutter filtering was used to suppress tissue clutter following the methods described in our previous fUS imaging study[48]. Power Doppler images were generated by integrating 250 frames. The fUS activation maps were then created based on Pearson's product-moment correlation coefficient between the stimulation pattern and the Power Doppler signals for each pixel, as described by ref. 32. For the whisker stimulation, facial whiskers were manually stimulated using a cotton swab. The stimulation pattern consisted of a total of three cycles, each of which included 30 s of stimulation and 30 s of rest.

After fUS imaging, fULM imaging of whisker stimulation was performed. Definity® MBs were diluted with saline and were infused in a continuous manner using a syringe pump with an injection rate of 60 $\mu L$/min. A magnet was placed within the syringe to mix the MB solution during the acquisition. The fULM data acquisition was performed using the same transducer and ultrasound system as the fUS experiment. Ultrasound data were obtained by using a 5-angle compounding plane-wave imaging sequence (step size of 2°) with a 6 V transmit voltage, center frequency of 15.625 MHz, PRF of 28.57 kHz, and post-compounding frame rate of 1000 Hz. IQ data of 800 frames per acquisition were acquired per second, and a total of 1050 acquisitions were generated (a total of 17.5 min of acquisition). Details of the in vivo data acquisition specifications and image resolution are summarized in Supplementary Table I.

Subpixel localizations were obtained using conventional localization or LOCA-ULM, and the localized centers were rounded to the chosen pixel size (9.856 $\mu m$ × 9.856 $\mu m$). Tracking of the localized centers was performed using uTrack, and the ULM images were reconstructed with a pixel size of 19.712 $\mu m$ × 19.712 $\mu m$ ($\lambda$/5 pixel size).

MB count map was computed as the total number of MBs for each pixel throughout the data acquisition period. fULM activation maps were generated based on Pearson's product-moment correlation coefficient $r$ between the stimulation pattern $A(t)$ and the MB count signal $s_{MB}(t)$ for each pixel described by ref. 32,

$$r = \frac{\sum_{i=1}^{N_t}\left(s_{MB}(t_i) - \hat{s}_{MB}(t)\right)\left(A(t_i) - \hat{A}\right)}{\sqrt{\sum_{i=1}^{N_t}\left(s_{MB}(t_i) - \hat{s}_{MB}(t)\right)^2}\sqrt{\sum_{i=1}^{N_t}\left(A(t_i) - \hat{A}\right)^2}}, \quad (12)$$

For calculating the activated pixel count, a correlation coefficient value above 0.2 was considered as activated pixels. The stimulation pattern consisted of a total of 15 cycles. Each cycle consisted of a 30-s stimulation period followed by a 40-second rest period. Specifically, the first 10 s of the rest period were a transition phase from stimulation to rest, and the subsequent 30 s were a stable rest phase (as illustrated in Fig. 6b). The fULM analysis focused on the stimulation periods (0 to 30 s within each cycle) and the stable rest periods (40 to 70 s within each cycle).

### Statistics and reproducibility
The study was designed to encompass a wide range of microbubble concentrations across both simulations and in vivo experiments. Preliminary pilot experiments ($n = 2$) for rat brain imaging were conducted for functional ULM protocol optimization purposes, beyond which no data were excluded. The experiments were not randomized. Data analysis was conducted and evaluated by computer algorithms, independent of human intervention. In this study, no grouping was applied, and the investigators were not blinded to allocation during experiments and outcome assessment.

### Reporting summary
Further information on research design is available in the Nature Portfolio Reporting Summary linked to this article.

## Data availability
The data for Figs. 3, 4, and 5 used in this study are available at Zenodo (DOI: 10.5281/zenodo.10711806)[49]. Data for Fig. 6 is available from the corresponding author upon request. Source data are provided with this paper.

## Code availability
The code to generate LSGAN-based microbubble signals and microbubble simulation pipeline is available at https://github.com/illyrs2/LOCA-ULM[50].

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

## Acknowledgements

This study was partially supported by the National Institute on Aging, the National Institute of Biomedical Imaging and Bioengineering, and the National Institute of Neurological Disorders and Stroke of the National Institutes of Health under grant numbers R21EB030072, R21EB030072-01S1, R21AG077173, R56NS131516, and by the National Science Foundation CAREER Award 2237166. The content is solely the responsibility of the authors and does not necessarily represent the official views of the NIH and NSF. MRL was supported by a Beckman Institute Postdoctoral Fellowship.

## Author contributions

Y.S. and P.S. designed and wrote the paper. Y.S. and X.C. designed the simulation study. Y.S., Y.W., Q.Y. and P.S. prepared the rat model and performed craniotomies and ultrasound imaging on rats. M.R.L. prepared the CAM model and performed ultrasound imaging. M.R.L. designed the noise model. Z.D. designed the ultrasound transducer holder and programmed the motorized imaging stage. Y.S., M.A.A. and P.S. developed and applied the super-resolution ULM algorithm.

## Competing interests

The authors declare no competing interests.
