## [Peer Review File · Nature Communications]

Context-Aware Deep Learning Enables High-Efficacy Localization of High Concentration Microbubbles for Super- Resolution Ultrasound Localization MicroscopyReviewer #1 (Remarks to the Author):

This manuscript studies deep learning-based ultrasound localization microscopy (ULM). The authors developed a generative adversarial network to model the complex characteristics of microbubbles (MBs) in vivo and generate realistic MB training data for the localization network with context awareness. The proposed LOCA-ULM localization pipeline showed its superior performance in MB detection accuracy, MB missing rate, and MB localization error compared to traditional ULM methods and other deep learning-based methods in the simulation study. In vivo CAM imaging and rat brain imaging study further validated its overall enhanced MB localization performance in different sized vessels and MB concentrations and faster and more robust vessel filling performance than compared methods, which could reduce both data acquisition time and post-processing time. Functional ULM experimental results demonstrated improved sensitivity to hemodynamics responses provided by LOCA-ULM, which suggests its great generalizability and application potential. The manuscript is well organized and the innovations are expressed clearly. I recommend publication after these issues are addressed.

1. The authors obtained poor localization performance when training DECODE with images generated by Gaussian PSF models. However, other deep learning-based localization networks like deep-ULM and mSPCN-ULM used the Gaussian PSF model to synthesize training data and the in vivo results seemed not that bad. The poor performance of DECODE trained by Gaussian PSF modeled training data may be caused by the architecture of the DECODE network rather than the low fidelity of training data. Additional experiments may be required to support the statement.
2. Only one comparison between the network trained with Gaussian PSF and that trained with LSGAN-generated PSF was undertaken in terms of ULM results of the rat brain. The performance of the network trained with Field-II PSF in the rat brain and the CAM microvessel model may be required for the comparison, as in the simulation study.
3. The authors convolved the ground truth positions with the synthesized MB PSFs randomly chosen from the MB PSF bank to generate simulated images. However, the MB positions in the imaging field posed a great impact on the MB PSF. Ignoring the variation in MB PSF due to the spatial location may cause the distortion of synthesized images. How these variations impact the accuracy of MB localization deserves investigation.
4. The generation of ground truth MB positions is not described clearly. Are there defined rules for generating the positions of the MBs? For example, do the authors employ a random generation process? Are there instances where multiple MBs are located in close proximity to each other, which potentially results in large image patches after convolution? Have the authors ensured the consistency of these defined rules in the comparative approaches?
5. How the imaging field of optical microscopy is aligned with the imaging field of ULM? How the vessels in the optical image are segmented? Whether the vessel segmentation results could be adopted as the ground truth of the CAM surface microvessels deserves further discussion.
6. How the integration of MB-specific characteristics including MB brightness levels, movement, lifetime, and ultrasound noise in the simulation framework improves the MB localization performance needs to be discussed in detail.
7. In Fig. 1 (e), it would be helpful to clarify what the yellow and red rectangles in the DECODE network represent.

Reviewer #2 (Remarks to the Author):

Summary of the paper

The manuscript by Shin et al. describes a novel deep learning method for Ultrasound Localization Microscopy (ULM) with improved performance both in silico and in vivo, which is highlighted by impressive results in functional ULM. Their contribution is two-fold. First, the authors propose a

novel approach to generate training data for deep-learning-based approaches based on PSF extracted in-vivo and synthetic PSF generated by an LS-GAN. Second, the authors translated the DECODE architecture introduced in SMLM into the ULM context. Their methodological improvements are backed with both in silico and in vivo results and are also validated against optical microscopy. Their results indicate that the proposed method leads to improved performance in the localization of microbubbles in ULM. These improvements also lead to a better sensibility in functional ULM.

Major Comments

- The results shown by the authors are convincing and show that the proposed approach indeed enhances ULM images. However, I believe they should be a bit more careful in determining why that is the case. For instance, on at least two occasions in the manuscript, the non-linear response of microbubbles is mentioned as a driving factor in the errors in PSF modeling from existing simulation software. I'm worried this claim is not fully supported by evidence. Indeed, we don't expect microbubbles to generate much non-linear responses at high frequencies, low MI and short exposure time as common in ULM. The authors rightfully identify other possible causes such as frequency-dependent attenuation, diffraction, phase aberration, multi-scattering, and multipath reverberation. Recent studies have shown that methods now exist to quantify some of these effects, for instance using the reflection matrix formalism (Lambert et al., Phys. Rev. X, 2020) while others have shown that degraded PSFs can be made to resemble simulated PSFs by applying phase correction (Demene et al., Nature BME, 2021, Robin et al., PMB, 2022, Xing et al., IEEE TMI, 2023). I think the authors might want to consider a different angle when describing how their approach can help with domain-shift problems in ULM. Indeed, describing it only as improved PSF modeling (and indirectly eliminating other important effects such as SVD residuals, large reflectors elsewhere in the image, etc.) might reduce the impact of their work while being somewhat inaccurate if we consider the definition of the PSF as being the image formed when there is a single scatterer (and nothing else) in the image.
- Another major concern about the approach to generating the training data as it is currently presented is that there is no guarantee that the subpixel position of the simulated scatterer exactly matches the simulated signal. Indeed, the simulation process is based on PSF extracted from in vivo data, and since the position of the scatterer is unknown, it is estimated using peak estimation based on a conventional approach. Consequently, especially in presence of large aberrations for instance in transcranial imaging, in vivo errors in localization of isolated scatterers from conventional methods could propagate in the training set.
- It is important to acknowledge that the improvement in architectures in LOCA-ULM is backed by comparison with several baseline methods (deep learning and conventional). However, the baseline simulation methods used to generate a comparable training set are limited to a Gaussian PSF convolution which is a very simple simulation model. The manuscript would benefit from a comparison like the LOCA-ULME vs LOCA-ULM (that shows the interest in using a GAN) but with a dataset generated, e.g., with Field II, to show the interest in using extracted PSF. In addition, it would be interesting and maybe easier to evaluate the performance of the method on a test set simulated with another model (e.g. Field II or SIMUS) to evaluate the propagation of the errors in the training set from the PSF extraction.
- The literature review does not cover several aspects of previous works related to the training data: the use of nonlinear simulations for deep learning model (Blanken et al. 2022), the prior use of Gan in ULM to obtain good performance in vivo at high concentration (Gu et al. 2023). In addition, it does not cover several prior works using the temporal context in the architecture of the network (Chen et al. 2022; Milecki et al. 2021, Xing et al., IEEE TMI, 2023). The manuscript would benefit from a more focused introduction on the problem addressed by the manuscript (i.e., producing better training data and improving the deep learning model).
- The computation performance subsection in the results is not convincing as it only relies on the time measurement of processing with fixed parameters. In addition, it is not clear if the LOCA-ULM and Conventional ULM computations are made on GPU or CPU. In the case where they are not performed on the same device, it should be specified as a limitation of the comparison and both CPU times could also be provided to provide some perspective.
- The authors need to better describe the tracking process along with a discussion on the potential biases it may cause. For instance, since LOCA-ULM is based on spatiotemporal datasets, it would be reasonable to expect longer track lengths. One could imagine obtaining longer track lengths

simply by temporally filtering datasets in conventional ULM, which may lead to an improved connection between vessels.

Specific comments

Line 20: 'revolutionized' is too strong

Line 25: missing word

Line 52: Claim is too strong – while it is true that resolution is better maintained with depth in ULM, a larger PSF leads, typically, to a poorer resolution in ULM

Line 83: SIMUS (Garcia, Comput Methods Programs Biomed, 2022) has also been used in deep-learning methods for ULM

Line 154: please clarify what 'real ultrasound data' means

Line 203: minimal resolution, not maximal

Line 278: It is unclear that the value in parenthesis (49.28 μ m) refers to the half wavelength and not to the wavelength.

Line 282: The sentence needs to be clarified.

Line 303: Please define what is meant by 'tiny' or remove

Line 331: The authors could consider broadening their claims to dynamic ULM methods in general and not only fULM.

Line 496: Please clarify the model used for the random fluctuations.

Reviewer #3 (Remarks to the Author):

comments in attach file

Please note that the Point Q7 was not fully shown in the file. Please see it below:

Q7 Line 234-245. Computational performance. Avoid repetitive superlatives about your method, and focus exclusively on computational time, not on repeating previously discussed advantages. The reader needs to understand for themselves that it is better. For example, "In addition to faster and more robust vessel filling performance" or « Collectively, these results indicate that not only does LOCA-ULM provide more robust MB localization »

There is no doubt that the LOCA-ULM can provide a faster algorithm, but the discussion about computation time is too simplistic. Computing time is highly dependent on the implementation of the algorithms. Please rewrite this section to include more useful information. What is the order of complexity of the algorithms (for example is an algorithm $O(2)$ or $O(3)$?), can you compare the total number of operations in both algorithms? Differences in the order of complexity of the algorithms or in the total number of operations can provide useful information about why your algorithm is better.

Are both algorithms implemented in GPU? please include this information in the main text.

Reviewer #3 Attachment on the following page

General Comment.

The authors propose a deep learning method to obtain ULM images with a higher density of microbubbles in the blood. The method is tested in the chicken embryo with a direct comparison between optical imaging and ULM image, in the rat brain to compare different bubble densities and separation methods, and finally applied to obtain functional images.

Although the proposed method is not a breakthrough, it shows interesting results in increasing bubble density, providing better or faster images and better functional images in a super-resolved modality.

However, the article needs a major revision to be more quantitative, to reduce some exaggerated claims and to focus attention on the real benefits of the method.

Major questions

Q1 General comment on superlatives and potentially exaggerated claims. The article uses too many superlatives, which reduces the overall quality of the article as we cannot distinguish a real improvement from an exaggerated claim. Please be as quantitative as possible. Revise the abstract and the end of the discussion. See also questions Q6 and Q7.

Q2 Figure 2.c and 2.d. The LOCA-ULM is compared with the Conventional ULM (no MB separation). This is not the best because the spectrum separation (Conventional MB separation) is better and has practically the same computational cost. Spectrum separation is a trivial procedure that enables to increase the number of detected bubbles you must always compare with a MB separation (see also Q6).

Q3 Figure 2.d In the image I estimate that the vessels have a minimum separation of $\sim 0.07\text{mm}$. At a relatively high frequency of 20MHz the wavelength is only 0.07mm, so this image does not show super resolution in the sense that we can separate two vessels below the diffraction limit. Of course, the vessels are highly contrasted and the diameters are very similar to the ground truth. Can you comment on that? Why did you use 20 MHz in this experiment and 15 MHz in the rat?

Q4 Line 171. LOCA-ULMe and LOCA-ULM are confusing. Can you clarify the methods? As it stands, I cannot see the interest in introducing the LOCA-ULMe, which is only proposed in the chicken embryo experiment.

Q5 line 247- 283. In this experiment the concentration is not explicit (main text or mat&met). This is important because you show later (Figure 4.j) that there is no benefit from using LOCA-ULM at low concentrations. I assume that the concentration is low (20ul/min?).

Q6 line 247- 283. The title of the section is "LOCA-ULM demonstrates superior in vivo ULM imaging performance in a rat brain", but at low concentrations both methods (LOCA-ULM and ULM with MB separation) are relatively equivalent. It is only at high concentrations that LOCA-ULM becomes interesting.

If we compare the LOCA-ULM with the conventional ULM (Fig. 3.c and d), we observe a clear gain in the LOCA-ULM, but if we compare it with the conventional WITH directional separation, this gain is relatively moderate. It is important to note that the directional separation is very simple (we split the spectrum in 2). For this reason, the comparison with the standard ULM without separation is of no interest, we must focus on the comparison with the state of the art, which is WITH separation.

The moderate gain of LOCA-ULM vs. ULM with separation is in agreement with figure 4.j. At low concentration, the conventional MB separation achieves to localise $6 \cdot 10^4$ bubbles compared to $8 \cdot 10^4$ with LOCA-ULM, which is a relatively moderate gain.

To avoid contradictions and to have a more synthetic view of the advantages of the method, the authors need to modify the figures and main messages of Figures 3 and 4. For example, they can simplify Figure 3 and use it as a proof of concept without major claims, and centre the gains of LOCA-ULM on the high concentrations (figure 4) or eliminate the no-MB-separate as they are not useful (you can always perform the spectral separation for free).

Q7 Line 234-245. Computational performance. Avoid repetitive superlatives about your method, and focus exclusively on computational time, not on repeating previously discussed advantages. The reader needs to understand for himself that it is better. For example, "In addition to faster and more

There is no doubt that the LOCA-ULM can provide a faster algorithm, but the discussion about computation time is too simple. Computing time is highly dependent on the implementation of the algorithms. Please rewrite this section to include more useful information. What is the order of the algorithms (n^2 n^3 ?), can you estimate the total number of operations in both algorithms? Differences in the order of the algorithms and in the total number of operations can give more information about why your algorithm is better. I see in the code examples that they run on GPU (is this the case for both algorithms?) Include this information in the main text.

Q8 Figures 4 and 5. The colors in Figures 4 and 5 are different from those in Figure 3. This makes it very difficult to compare the figures. Please use a single color code for both figures, it is also appreciated to include a detail of the cortical part in all cases as in figure 3.

Q9 Figure 4. Why do you stop the concentration at 40ul/min in this experiment? In the fUS experiment the concentration is increased to 60ul/min. Do you reach saturation of the LOCA-ULM at 60ul/min?

Q10 Lines 320-330 figure 6. Two other methods are presented here (Deep-ULM and mSPCN) but there is no bibliography and they are not mentioned in the methods section. Include the concentration in the main text (the 40ul/min is only in the caption).

Q11 Lines 331-378. The functional experiment is an interesting result. Can you compare it to a standard correlation with power Doppler (if you did the experiment)? In my opinion, both methods are complementary, standard FUS could be more sensitive and identify the regions without ambiguity. Outside the activated regions, the LOCA-ULM image has a lot of negative correlations and also mixed positive and negative correlations that are difficult to analyse. Are these significant or random noise? You can add a supplementary figure showing only the significant activated pixels. Please also explicitly mention in the main text that the concentration (60ul/min) was higher than the previous ones.

Q12 Methods line 626. The chicken embryo experiment is very interesting. It allows a direct comparison between the optical image (a ground truth) and the ULM image (Fig. 2.e). The description of this experiment is minimal in the methods. Since the imaging planes in optics and ultrasound are orthogonal (optics sees the surface, ultrasound cuts the medium), how do you place the ultrasound probe in the same plane as the optical image? Please provide a detailed description of this experiment.

o

r

«

»

Q13 Methods. The different experiments use different parameters, bubble concentrations and ultrasound parameters (1° steps or 2° steps, 5 or 9 angles, 15MHz and 20MHz, the voltage is only given in the fUS 6V) Can you summarise all the parameters in the supplementary table?

Minor questions

Q14 Figure 1. The legend inside panel e (DECODE Network) is not visible and not explained in the caption or in the text.

Q15 Line 132. The acronym CAM is not specified in the text.

Q16 line 284. “Our result suggests that the higher number of microvessels revealed by LOCA-ULM is indeed from localizations of real MBs instead of false ones”. This is a hypothesis, but it is difficult to verify.

Q17 Figure 6. Can you overlay the atlas to show that the activated regions are actually S1BF and VPM (a simple line might suffice).

Q18 Figure 6 Can you indicate the position of the imaged plane relative to the standard bregma point? (idem in the other figures)

Q19 The voltage is only 6V (in the fUS experiment). Higher voltages destroy the bubbles?

Q20 There is no mention of the ultrasound electronics and software in the methods (I assume a Verasonics?).

Q21 Supplementary table. The 15.625Mhz of the L22-14vX probe is probably not the central frequency. It's probably the demodulation of the Verasonics electronics (it samples at 4x15.625Mhz). Check the central frequency in the probe calibration.

Reviewer #1 (Remarks to the Author):

This manuscript studies deep learning-based ultrasound localization microscopy (ULM). The authors developed a generative adversarial network to model the complex characteristics of microbubbles (MBs) *in vivo* and generate realistic MB training data for the localization network with context awareness. The proposed LOCA-ULM localization pipeline showed its superior performance in MB detection accuracy, MB missing rate, and MB localization error compared to traditional ULM methods and other deep learning-based methods in the simulation study. *In vivo* CAM imaging and rat brain imaging study further validated its overall enhanced MB localization performance in different sized vessels and MB concentrations and faster and more robust vessel filing performance than compared methods, which could reduce both data acquisition time and post-processing time. Functional ULM experimental results demonstrated improved sensitivity to hemodynamics responses provided by LOCA-ULM, which suggests its great generalizability and application potential. The manuscript is well organized, and the innovations are expressed clearly. I recommend publication after these issues are addressed.

We thank the reviewer for their positive and insightful comments. Below we have provided detailed responses addressing the comments raised by the reviewer.

R1.1) The authors obtained poor localization performance when training DECODE with images generated by Gaussian PSF models. However, other deep learning-based localization networks like deep-ULM and mSPCN-ULM used the Gaussian PSF model to synthesize training data and the *in vivo* results seemed not that bad. The poor performance of DECODE trained by Gaussian PSF modeled training data may be caused by the architecture of the DECODE network rather than the low fidelity of training data. Additional experiments may be required to support the statement.

Response Thank you for pointing this out. We have conducted a comparative analysis of Deep-ULM, mSPCN, and LOCA-ULM, with each model trained using Gaussian MB templates. As shown in Supplementary Fig. 6 (also presented below for the reviewer's convenience), the DECODE network experienced a decrease in localization performance when trained with Gaussian MB signals, which do not accurately represent real MB signals. A similar decline in localization performance was observed in both Deep-ULM and mSPCN. This suggests that these issues may not be unique to the DECODE architecture: the domain mismatch caused by MB modeling universally impacts the performance of deep learning-based localization methods. These findings are consistent with recent literature report presenting a similar drop in localization performance with Deep-ULM and mSPCN (Hahne C. et al., *arXiv preprint [v1]* (2023)). Nevertheless, LOCA-ULM trained with either Gaussian or LSGAN MB signals still demonstrated better localization performance than both Deep-ULM and mSPCN (as shown in Supplementary Fig. 6), indicating the enhanced localization performance of the DECODE network.

The Supplementary Figures and the Discussion section have been updated accordingly to reflect the reviewer's comment:

Page 19 Line 461 When trained with Gaussian MB templates, all the deep learning-based MB localization methods (Deep-ULM, mSPCN, and LOCA-ULM) exhibited suboptimal localization performance, which manifested as the presence of gridding artifacts and poor vessel reconstructions (Supplementary Fig. 6).

Supplementary Fig 6. Comparison of Deep-ULM, mSPCN, and LOCA-ULM trained with simulation data generated using either Gaussian- or LSGAN-based MB templates. *In vivo* rat brain imaging data were used for testing.

- Hahne, Christopher, et al. "RF-ULM: Deep Learning for Radio-Frequency Ultrasound Localization Microscopy." *arXiv preprint arXiv:2310.01545* (2023) [v1] (<https://arxiv.org/pdf/2310.01545v1.pdf>).

R1.2) Only one comparison between the network trained with Gaussian PSF and that trained with LSGAN-generated PSF was undertaken in terms of ULM results of the rat brain. The performance of the network trained with Field-II PSF in the rat brain and the CAM microvessel model may be required for the comparison, as in the simulation study.

Response We appreciate the reviewer's suggestion for a more comprehensive comparison based on using different MB simulation models. In Supplementary Fig. 7 (presented below), we have expanded our

comparison study to include results from using all three different MB models (Gaussian, Field II, and LSGAN) in both CAM and rat brain study. As can be seen in these new results, LOCA-ULM trained with LSGAN-generated MB templates consistently outperformed those trained with Gaussian and Field II MB templates. We have added these new results in Supplementary Figures and the Discussion section:

Page 19 Line 464 However, when LOCA-ULM was trained with MB templates generated by LSGAN, it demonstrated enhanced ULM reconstruction over those trained by Gaussian and Field-II MB templates. This improvement is characterized by enhanced contrast, reduced gridding artifacts, and a more comprehensive reconstruction of vascular structures for the chicken embryo chorioallantoic membrane (CAM) and rat brain (Supplementary Fig. 7).

Supplementary Fig 7. Comparison of reconstructed ULM images using different MB simulation methods: Gaussian, Field II, and LSGAN. a ULM imaging results in the CAM study, **b** ULM imaging results in the rat brain study. The ULM images demonstrate the impact of MB model selection on image quality, contrast, and artifact presence in ULM. Scalebars represent 1 mm.

R1.3) The authors convolved the ground truth positions with the synthesized MB PSFs randomly chosen from the MB PSF bank to generate simulated images. However, the MB positions in the imaging field posed a great impact on the MB PSF. Ignoring the variation in MB PSF due to the spatial location may cause the distortion of synthesized images. How these variations impact the accuracy of MB localization deserves investigation.

Response We acknowledge the reviewer's concerns regarding the potential impact of spatial variations of MB PSFs on localization accuracy. In response, we have conducted a detailed analysis and new experiments that will be discussed below. Before presenting the new results, it is important to clarify that convolutional neural networks (CNNs) are spatially invariant and exhibit robustness against variations in PSFs. This robustness persists provided that two conditions are met: the PSF variations between different field-of-views (FOVs) are not substantial, and the training data encompasses the entire distribution of MB PSFs observed *in vivo*. The processing steps within convolutional layers (e.g., convolution and pooling) are designed to learn different features from various spatial locations. These features are then embedded into the latent space, meaning they can handle spatial variations. Therefore, LOCA-ULM was trained with MB signals extracted from the entire ultrasound imaging FOV to achieve robust localization performance across different regions.

With that in mind, we would like to introduce some new experiment results to evaluate the MB signal variations across different regions of the imaging FOV. We used materials with realistic tissue attenuation to create an imaging environment where the frequency-dependent-attenuation-induced MB PSF variation as observed *in vivo* could be emulated. We used a 60:40 volume ratio mixture of condensed milk and water to create a solution that: 1) has similar acoustic attenuation properties to brain tissue, specifically 0.50 dB/cm/MHz (Duck, F. Academic press (2013), Aubry, J. F., et al., arXiv preprint (2023)); 2) provides a liquid environment to allow even distributions of MBs at different imaging depths.

The attenuation coefficient was measured using a simple experimental setup as illustrated in Supplementary Fig. 8a below, where two identical CTS 10MHz IS1004HR transducers were positioned 5.5 cm apart to operate in a “pitch-and-catch” mode to assess ultrasound signal attenuation through propagation in the sample material contained in the testing cell. An ultrasound pulser (Olympus 5800 PR) was used to drive the transmitting transducer and a DAQ (NI PXI5124) was used to sample the ultrasound signal acquired at the receiving transducer. The attenuation coefficient α was calculated by:

$$\alpha = \frac{1}{f} \frac{20}{d} \log_{10} \left(\frac{A_0}{A} \right)$$

where d is the propagation distance (i.e., 5.5 cm), f is the ultrasound frequency (10 MHz), and A_0 , A are the amplitude corresponding to the peak of the received signal for pure water and condensed milk-water mixture, respectively. The experiment was conducted twice, yielding an attenuation coefficient of approximately 0.50 dB/cm/MHz and 0.51 dB/cm/MHz, which is similar to the attenuation coefficient of brain tissue (Duck, F. Academic press, (2013), Aubry, J. F., et al., arXiv preprint (2023)).

After validating the attenuation coefficient, activated Definity MBs were diluted by a factor of 5000 in distilled water. 10 μ L of this diluted solution was carefully mixed with degassed condensed milk-water mixture, resulting in a final volume of 100 mL. We then acquired experimental MB data using the same imaging setup as the *in vivo* rat brain imaging study. To analyze the MB signals, we divided the FOV into three regions, corresponding to three depth segments: 2-5 mm, 5-8 mm, and 8-11 mm (Supplementary Fig. 8b). A total of 3780 individual MB signals were extracted across all imaging zones followed by full width

at half maximum (FWHM) measurements (in both the axial and lateral dimensions) for the analysis of MB signal distribution.

In Supplementary Fig. 8c, the axial FWHM of experimental MBs varies from 134.97 μm to 147.27 μm , while the lateral FWHM varies from 192.12 μm to 247.87 μm from the shallowest to the deepest regions. We subsequently trained the LSGAN model using 3780 experimental MB signals and generated an equal number of synthetic MB signals. As shown in Supplementary Fig. 8d, the distribution of LSGAN-generated MBs closely follows the distribution of the experimental MBs, as evidenced by the similar average lateral and axial FWHM values for LSGAN-generated MBs ($\mu_x = 215.48 \mu\text{m}$, $\mu_z = 144.87 \mu\text{m}$) and experimental MBs ($\mu_x = 218.69 \mu\text{m}$, $\mu_z = 144.34 \mu\text{m}$). Through this experiment, we show that the LSGAN-generated MB templates encompass the MB signal distribution across the entire ultrasound FOV, thereby enhancing the robustness of LOCA-ULM. The following results has been added to the Supplementary Information and Discussion.

Page 19 Line 470 However, it is important to consider the spatial variations of MB signals and their potential impact on localization accuracy. As illustrated in Supplementary Fig. 8c, our brain-tissue mimicking phantom experiment indicates that MB signals do present depth-dependent variations (Supplementary Methods 4). However, the MB templates generated by LSGANs follow a similar distribution as experimental MB signals across the entire FOV (Supplementary Fig. 8d). Therefore, the training data using LSGAN-generated MBs does include a complete MB distribution from different imaging depths, facilitating a robust MB localization performance for LOCA-ULM. It should be noted that the performance on other transducer arrangements (e.g., curvilinear) was not investigated in this study.

Supplementary Fig 8. Experiment setup for attenuation coefficient measurement and experimental MB signal analysis **a** A custom-built housing with a testing cell sandwiched between two CTS 10MHz IS1004HR transducers operating in a “pitch-and-catch” mode. A mixture of condensed milk and water was used as the testing medium to simulate rat brain tissue. **b** B-mode ultrasound image displaying MBs suspended within the testing medium. White dashed lines denote the three analyzed depth zones. **c** Histograms showing the full width at half maximum (FWHM) of MB signals in lateral (blue) and axial (green) dimensions across different depths. **d** Histograms showing the FWHM for experimental MB signals (across entire FOV) and LSGAN-generated MB signals.

- Duck, Francis. *Physical properties of tissues: a comprehensive reference book*. Academic press, 2013.
- Aubry, Jean-Francois, et al. "ITRUSST Consensus on Biophysical Safety for Transcranial Ultrasonic Stimulation." *arXiv preprint arXiv:2311.05359* (2023).

R1.4) The generation of ground truth MB positions is not described clearly. Are there defined rules for generating the positions of the MBs? For example, do the authors employ a random generation process? Are there instances where multiple MBs are located in close proximity to each other, which potentially results in large image patches after convolution? Have the authors ensured the consistency of these defined rules in the comparative approaches?

Response Thank you for your feedback. We have indeed established rules for creating MB positions and trajectories. These rules start with randomly determined initial positions, followed by a motion model that introduces random variations in velocity. We have also taken note of instances where MBs are positioned closely. The same MB position generation pipeline was used to generate the training datasets using different

MB templates (i.e., Gaussian, Field II, and LSGAN) and network architectures (i.e., Deep-ULM, mSPCN, and LOCA-ULM). Below, we provide clearer explanations and the updates made to our Methods section:

MB motion model We have added a detailed explanation of the process for defining MB positions in our simulation. Initially, each MB is assigned with a constant speed and a two-dimensional (2D) unit vector representing its direction. To simulate stochastic MB behavior, small random 2D vectors are added as perturbations to the direction vector at each time step. This leads to a random variation in velocity, resulting from the product of constant speed and the perturbed direction vector.

Initialization of MBs: In the revised methods, we specified that the initial positions of MBs, represented by (x,y) coordinates, were sampled from random locations within the 2D FOV.

Speed: The speed of each MB was determined by randomly sampling from a uniform distribution that ranges between 5 mm/s and 25 mm/s.

Direction: The 2D unit vector, which represents the direction of an MB, is perturbed at each time step by adding a small 2D random vector at each time step. The components of this random vector are sampled from a normal distribution with a standard deviation of 0.2.

Velocity: After the random direction perturbation, the direction vector was multiplied by the constant speed to derive the velocity.

Page 23 Line 570 In our simulation pipeline, we modeled MB flow within vessels using a motion model that accounts for random changes in MB movement. The initial positions of MBs were randomly sampled from a uniform distribution within the dimensions of the imaging FOV. The lifetime of each MB ranged from 1 to 20 frames, randomly sampled from a uniform distribution. Each MB is assigned an initial speed and a two-dimensional (2D) unit vector that represents the initial direction. The speed was randomly sampled from a uniform distribution between 5 mm/s and 25 mm/s. To simulate stochastic MB motion, the direction of the MBs was perturbed at every time step by adding a small random vector. The components of this random vector were sampled from a normal distribution with a standard deviation of 0.2. After the random direction perturbation, the direction vector and the speed are multiplied to obtain the velocity. The MB position was updated according to the velocity, and the position at each time frame was stored as the ground truth.

As shown in the figures below, our simulation did include sample cases with closely distributed MBs. Our data generation process was robust enough to accommodate high MB densities and generate realistic MB images.

Fig. R1 Examples of simulated MB images using Field II and LSGAN-generated MB signals. Red dots indicate the ground truth MB location.

Finally, we used the same simulation pipeline for training data generation across different MB template generation (Gaussian, Field II, and LSGAN) and different DL networks (DeepULM, mSPCN, and LOCA-ULM), ensuring a standardized simulation and training process. This has been clarified in the Supplementary Information.

Supplementary Methods 3. Deep-ULM and mSPCN Implementations

The training datasets for LOCA-ULM, Deep-ULM, and mSPCN were generated using the same MB flow simulation (motion model, brightness, lifetime, and velocity) and MB templates (i.e., LSGAN, Gaussian, and Field II MB templates).

R1.5) How the imaging field of optical microscopy is aligned with the imaging field of ULM? How the vessels in the optical image are segmented? Whether the vessel segmentation results could be adopted as the ground truth of the CAM surface microvessels deserves further discussion.

Response The reviewer posed an important question regarding the alignment of optical images with ultrasound and the image segmentation process. First, we have introduced new discussions about how CAM optical images are widely adopted as the reference for ultrasound CAM surface microvessel imaging.

Page 30 Line 720 For our study, we used the CAM microvessel model and optical imaging to provide the reference ground truth for validating different MB localization methods. The *ex ovo* CAM microvessels present a dense and diverse network of blood vessels with diameters ranging from 10 to 155 μm ⁴¹. The optical transparency of CAM allows for robust registration between ultrasound and optical images and is therefore widely used as a benchmark for assessing microvasculature with contrast-enhanced ultrasound^{10, 42-44}.

Regarding the optical imaging segmentation process, the task was indeed challenging and required a large amount of experimental effort and experience to accomplish. A detailed description is provided below which has been added to the revised manuscript. We would like to point out that despite the effort, misalignment and mis-segmentation can still occur due to the considerable differences in the imaging setup, instrumentation, and imaging physics between optical and ultrasound imaging. We also note that the segmentation process was not trivial on CAM images. We have experimented with a wide variety of segmentation solutions but eventually resorted to manual segmentation to account for the complex vascular structures *in vivo*. Nevertheless, since the *same* optical images (and the segmented results) were used as a benchmark across all ultrasound imaging methods, our study design is still valid, and the comparison results are still meaningful.

Page 28 Line 685 First, a region of interest (ROI) was carefully selected to include matching vascular structures in both imaging modalities. For optical microscopy images, vascular structures were manually segmented and used as the reference. This manual segmentation was executed using the GNU Image Manipulation Program (GIMP). The resulting optical binary map was then spatially registered with the reconstructed ULM image, which involved resizing and cropping for proper alignment.

R1.6) How the integration of MB-specific characteristics including MB brightness levels, movement, lifetime, and ultrasound noise in the simulation framework improves the MB localization performance needs to be discussed in detail.

Response We appreciate the opportunity to elaborate on the impact of MB-specific characteristics within our simulation framework. We have conducted an ablation study, where we systematically excluded MB

brightness, movement and lifetime, and background noise to determine how each factor individually impacted the performance of LOCA-ULM

- **MB Brightness:** Removing MB brightness (i.e., applying a constant brightness level to all simulated MBs) resulted in a marked decline in detection efficiency (see Supplementary Fig. 3b below). This highlights the importance of accounting for brightness variability, as a constant brightness does not reflect the true range of MB intensities, resulting in missed detections (green arrows in Supplementary Fig. 3b).
- **MB movement and Lifetime:** The exclusion of MB velocity and lifetime also undermined the MB localization performance, particularly in areas of high MB density (yellow arrow in Supplementary Fig. 3c). Although the final reconstructed ULM image was not as compromised as without MB brightness, the larger vessel area presents a considerable drop in intensity (blue arrow in Supplementary Fig. 3c), indicating a decrease in MB localization efficacy. Future studies will input a larger number of frames into the network, aiming to better utilize temporal information for improved localization performance.
- **Background Noise:** The absence of simulated background noise led to misclassification of noise as MBs, resulting in increased false positives (red arrows in Supplementary Fig. 3d).

We have updated the Supplementary Figures and the Results section of the manuscript to reflect these findings.

Supplementary Fig 3. Ablation study of MB signal characteristics in the LOCA-ULM simulation framework. Top row: **a** ULM images with all MB characteristics included (All), **b** without brightness variations (w/o Brightness), **c** without MB movement and lifetime (w/o Lifetime/Velocity), and **d** without simulated background ultrasound noise (w/o Background Noise). Bottom row: *in vivo* rat brain contrast-enhanced B-mode image and localization results under each ablation scenario.

Page 14 Line 329 *MB specific characteristics improve the localization performance*

An ablation study was conducted to evaluate the impact of MB-specific characteristics on MB localization performance within the LOCA-ULM simulation framework. Each MB characteristic used in this study,

including MB brightness levels, MB movement and lifetime, and background ultrasound noise, was individually excluded from the framework to determine their impact on MB localization. As shown in Supplementary Fig. 3b, the absence of MB brightness variations resulted in a marked reduction in MB detection rate (Supplementary Fig. 3b green arrows), resulting in an incomplete ULM reconstruction. On the other hand, the exclusion of MB movement and lifetime mostly impacted the larger vessel regions with higher MB densities (Supplementary Fig. 3c, yellow and blue arrows). Finally, the absence of background noise led to an increase in false-positive MB localizations because a network trained without noise tends to misclassify background noise as MBs (Supplementary Fig. 3d, red arrows). These results underscore the importance of integrating MB characteristics into the training pipeline to introduce context-dependent features to the network, thereby mitigating the domain mismatch between training and testing datasets to facilitate high-fidelity ULM reconstructions.

R1.7) In Fig. 1 (e), it would be helpful to clarify what the yellow and red rectangles in the DECODE network represent.

Response We provided additional clarifications in the figure legend. The red rectangles in the DECODE network diagram represent the output layers, while the yellow rectangles denote the layers that are skip-connected, as indicated by the blue arrows. We discarded the description of the yellow rectangles to avoid redundancy.

Reviewer #2 (Remarks to the Author):

Summary of the paper

The manuscript by Shin et al. describes a novel deep learning method for Ultrasound Localization Microscopy (ULM) with improved performance both in *silico* and in *vivo*, which is highlighted by impressive results in functional ULM. Their contribution is two-fold. First, the authors propose a novel approach to generate training data for deep-learning-based approaches based on PSF extracted in-*vivo* and synthetic PSF generated by an LS-GAN. Second, the authors translated the DECODE architecture introduced in SMLM into the ULM context. Their methodological improvements are backed with both in *silico* and in *vivo* results and are also validated against optical microscopy. Their results indicate that the proposed method leads to improved performance in the localization of microbubbles in ULM. These improvements also lead to a better sensibility in functional ULM.

We thank the reviewer for their thorough examination of the manuscript and insightful feedback, which led to significant improvement of the manuscript. Below we have provided detailed responses to all the comments raised by the reviewer.

Major Comments

R2.1) The results shown by the authors are convincing and show that the proposed approach indeed enhances ULM images. However, I believe they should be a bit more careful in determining why that is the case. For instance, on at least two occasions in the manuscript, the non-linear response of microbubbles is mentioned as a driving factor in the errors in PSF modeling from existing simulation software. I'm worried this claim is not fully supported by evidence. Indeed, we don't expect microbubbles to generate much non-linear responses at high frequencies, low MI and short exposure time as common in ULM. The authors rightfully identify other possible causes such as frequency-dependent attenuation, diffraction, phase aberration, multi-scattering, and multipath reverberation. Recent studies have shown that methods now exist to quantify some of these effects, for instance using the reflection matrix formalism (Lambert et al., Phys. Rev. X, 2020) while others have shown that degraded PSFs can be made to resemble simulated PSFs by applying phase correction (Demene et al., Nature BME, 2021, Robin et al., PMB, 2022, Xing et al., IEEE TMI, 2023). I think the authors might want to consider a different angle when describing how their approach can help with domain-shift problems in ULM. Indeed, describing it only as improved PSF modeling (and indirectly eliminating other important effects such as SVD residuals, large reflectors elsewhere in the image, etc.) might reduce the impact of their work while being somewhat inaccurate if we consider the definition of the PSF as being the image formed when there is a single scatterer (and nothing else) in the image.

Response We sincerely appreciate the insightful feedback provided by the reviewer, prompting us to reevaluate the definition of PSF modeling and the LOCA-ULM simulation framework.

The PSF is a measure of how an ultrasound system represents a point target, essentially serving as the spatial impulse response of the system (Zemp, R. J., et al., IEEE TUFFC, 2003). In this paper, we consider the PSF in the broadest sense where it encompasses all the components involved with microbubble imaging, including transmit and receive beamforming, acoustic wave propagation (both linear and nonlinear propagation, attenuation, reverberation, multi-scattering, aberration, etc. (G. F. Pinton, et al., IEEE TUFFC, 2011)), the imaging system (e.g., hardware components such as transducers and system circuitry), imaging settings (e.g., TGC), MB acoustic responses (M. Versluis, et al., Ultrasound in Medicine & Biology, 2020), tissue properties (e.g., different types of tissues and blood flow conditions), and the postprocessing components (e.g., clutter filtering, denoising, etc.).

As such, *in vivo* PSF estimation or modeling is a challenging task which involves inherent uncertainties due to the complex ultrasound image formation process, leading to inevitable domain mismatches between real ultrasound data and training data. We agree with the reviewer that MB nonlinear responses may only be one of the components that led to domain mismatch, and it is indeed important to deliver the appropriate perspectives on LOCA-ULM, which used a data-driven approach to address the domain-specific complexities encountered in *in vivo* contrast-enhanced ultrasound imaging. In our revised manuscript, we emphasize that LOCA-ULM operates primarily as an MB template generator serving two main objectives: (1) generating realistic MB signals to minimize the domain mismatch between training and testing data; and (2) offering a high degree of flexibility in creating diverse MB signals to enhance the network's generalization capability for MB localization.

The following content has been added to the revised manuscript to discuss this point:

Page 4 Line 84 However, these simulations fall short of creating complex and spatiotemporally varying MB signals observed *in vivo*, which can be attributed to numerous factors involved in the MB imaging process. These factors include transmit and receive beamforming, acoustic wave propagation (both linear and nonlinear propagation, attenuation, reverberation, multi-scattering, aberration, etc.¹⁷), the imaging system (e.g., hardware components such as transducers and system circuitry), imaging settings (e.g., TGC), MB acoustic responses¹⁸, tissue properties (e.g., different types of tissues and blood flow conditions), and the postprocessing components (e.g., clutter filtering, denoising, etc.).

Page 4 Line 102 One main contribution of LOCA-ULM was overcoming the difficulties of generating realistic MB training data for developing deep learning-based localization techniques for *in vivo* contrast-enhanced ultrasound imaging applications. The proposed generative adversarial network (GAN)²⁰-based architecture was able to learn the distribution of real MB signals acquired *in vivo* and generate diverse MB templates that are essential for developing generalizable solutions for a variety of imaging conditions with different types of biological tissues. To further minimize the mismatch between the simulation and real data, we incorporated ultrasound system noise modeling and key MB attributes (e.g., brightness levels, lifetime, and movement velocity) into the simulation process. Collectively, LOCA-ULM demonstrated marked improvement in MB detection accuracy and presented a practical solution to solving the domain discrepancy problem when developing deep learning-based imaging techniques that involve the use of MBs.

- Zemp, Roger J., Craig K. Abbey, and Michael F. Insana. "Linear system models for ultrasonic imaging: Application to signal statistics." *IEEE transactions on ultrasonics, ferroelectrics, and frequency control* 50.6 (2003): 642-654.
- Pinton, Gianmarco F., Gregg E. Trahey, and Jeremy J. Dahl. "Sources of image degradation in fundamental and harmonic ultrasound imaging using nonlinear, full-wave simulations." *IEEE transactions on ultrasonics, ferroelectrics, and frequency control* 58.4 (2011): 754-765.
- Versluis, Michel, et al. "Ultrasound contrast agent modeling: a review." *Ultrasound in medicine & biology* 46.9 (2020): 2117-2144.
- Lambert, William, et al. "Reflection matrix approach for quantitative imaging of scattering media." *Physical Review X* 10.2 (2020): 021048.
- Blanken, Nathan, et al. "Super-resolved microbubble localization in single-channel ultrasound RF signals using deep learning." *IEEE transactions on medical imaging* 41.9 (2022): 2532-2542

R2.2) Another major concerns about the approach to generating the training data as it is currently presented is that there is no guarantee that the subpixel position of the simulated scatterer exactly matches the simulated signal. Indeed, the simulation process is based on PSF extracted from *in vivo* data, and since the position of the scatterer is unknown, it is estimated using peak estimation based on a conventional approach. Consequently, especially in presence of large aberrations for instance in transcranial imaging, *in vivo* errors in localization of isolated scatterers from conventional methods could propagate in the training set.

Response Thank you for your important feedback. This is indeed a good point, and we acknowledge that using MB templates extracted from *in vivo* data carries inherent uncertainties of MB position into the training data. This is further addressed in our response to Comment 3 below (R2.3), where we have outlined the additional experiments conducted using the Field II simulation to quantify the propagation of errors.

R2.3) It is important to acknowledge that the improvement in architectures in LOCA-ULM is backed by comparison with several baseline methods (deep learning and conventional). However, the baseline simulation methods used to generate a comparable training set are limited to a Gaussian PSF convolution which is a very simple simulation model. The manuscript would benefit from a comparison like the LOCA-ULME vs LOCA-ULM (that shows the interest in using a GAN) but with a dataset generated, e.g., with Field II, to show the interest in using extracted PSF. In addition, it would be interesting and maybe easier to evaluate the performance of the method on a test set simulated with another model (e.g., Field II or SIMUS) to evaluate the propagation of the errors in the training set from the PSF extraction.

Response To address Comment 2 and 3, we conducted an additional experiment using Field II simulation. This experiment allowed us to examine the propagation of MB localization errors that originated from *in vivo* MB template extraction using conventional localization. Specifically, we simulated phase aberrations in individual MBs by introducing perturbations to the phase/speed of sound in the RF signal (examples are shown in Fig. R2, Methods are detailed below). LOCA-ULM was trained using the phase-distorted Field II MB templates, which were extracted using conventional localization based on normalized cross-correlation. As shown in Supplementary Fig. 1, the mean lateral and axial localization errors for LOCA-ULM ($\mu_x = 0.412 \mu m$, $\mu_z = -8.840 \mu m$) align with those from conventional localization ($\mu_x = 0.348 \mu m$, $\mu_z = -8.388 \mu m$).

The accuracy of localizing isolated MBs pertains to determining their true physical locations, which are unattainable *in vivo*. The results above suggest that the accuracy of localizing isolated MBs for LOCA-ULM is inherently limited by the conventional localization (as correctly pointed out by the reviewer), which serves as the basis for our ground truth. Therefore, correcting distortions of the PSF (e.g., using phase aberration correction) before localization is essential to enhance localization accuracy. However, the accuracy of localizing isolated MBs does not directly correspond to the effectiveness of LOCA-ULM in identifying MB centers within clusters of MBs. The capability of LOCA-ULM to differentiate MBs in overlapping MB distributions is what distinguishes our method from the existing ones, as evidenced by the substantially improved ULM reconstruction quality in this paper. We have included detailed results and discussions of these findings in the revised manuscript and the Supplementary methods.

Fig. R2 Examples of simulated MB images using Field II with and without phase aberration.

Supplementary Fig 1. Comparative analysis of LOCA-ULM and conventional localization methods using a Field II simulated test set with induced phase aberration. The histograms show the distribution of the lateral and axial localization errors respect to the ground truth positions.

Page 12 Line 279 Considerations: Evaluating the Impact of Conventional Localization Error on LOCA-ULM Precision

LOCA-ULM demonstrated superior localization precision at high MB concentrations in our experiments. However, it is important to note that the simulation pipeline for LOCA-ULM used MB templates extracted from *in vivo* data, with peaks identified by the conventional localization method serving as the ground truth. Therefore, any MB localization errors generated from using conventional localization can propagate into the LOCA-ULM training dataset. To quantify the error propagation, we trained LOCA-ULM using Field

II-simulated MB templates that included phase aberration errors (Supplementary Methods 1). Supplementary Fig. 1 displays the localization errors for both LOCA-ULM and conventional localization. The mean lateral and axial localization errors for LOCA-ULM ($\mu_x = 0.412 \mu\text{m}$, $\mu_z = -8.840 \mu\text{m}$) align closely with those for conventional localization method ($\mu_x = 0.348 \mu\text{m}$, $\mu_z = -8.388 \mu\text{m}$). These findings suggest that the accuracy of LOCA-ULM in localizing isolated MBs is inherently limited by the ground truth localization estimate in the training phase. Thus, addressing PSF distortions (e.g., by phase aberration correction or other image-quality-enhancing beamforming methods) before localization remains an essential step to improving localization accuracy.

Supplementary Methods 1 Field II phase-aberrated microbubble (MB) simulation

To study the effect of propagation of localization error from conventional localization, we simulated 200 images of phase-aberrated MBs using Field II. The Field II simulation sequence was configured according to the CAM imaging settings described in Supplementary Table 1. In each image, single MBs were randomly positioned within a 2D field that was 10 mm wide, extending from $z = 2$ mm to $z = 15$ mm in depth, and with a pixel size of 0.064λ . We simulated phase aberrations in individual MBs by introducing perturbations to the phase/speed of sound in the RF signal. Random phase delays for each element were sampled from a uniform distribution between 0 and π radians. The corresponding time delay based on the central frequency (20 MHz) was then applied to the transmit and receive delay for each element. We used conventional localization to extract Field II MB templates for training the LOCA-ULM network. The localization accuracy of both the conventional method and LOCA-ULM was compared against the ground truth positions (Supplementary Fig. 1).

Moreover, in response to the similar comments raised by Reviewer 1 Comment 2 (**R1.2**), we have conducted a more comprehensive comparison based on using different MB simulation models (Gaussian, Field II, and LSGAN). In Supplementary Fig. 7 (copied below for the convenience of the reviewer), we have expanded our analysis to include the LOCA-ULM performance trained with Gaussian, Field II, and LSGAN-generated MB signals for CAM and rat brain imaging study. Our new results demonstrate that LOCA-ULM trained with LSGAN-generated MB templates consistently outperformed those trained with Gaussian and Field II MB templates.

Page 19 Line 464 However, when LOCA-ULM was trained with MB templates generated by LSGAN, it demonstrated enhanced ULM reconstruction over those trained by Gaussian and Field-II MB templates. This improvement is characterized by enhanced contrast, reduced gridding artifacts, and a more comprehensive reconstruction of vascular structures for the chicken embryo chorioallantoic membrane (CAM) and rat brain (Supplementary Fig. 7).

Supplementary Fig 7. Comparison of reconstructed ULM images using different MB simulation methods: Gaussian, Field II, and LSGAN. a ULM imaging results in the CAM study, **b** ULM imaging results in the rat brain study. The ULM images demonstrate the impact of MB model selection on image quality, contrast, and artifact presence in ULM. Scalebar represents 1 mm.

R2.4) The literature review does not cover several aspects of previous works related to the training data: the use of nonlinear simulations for deep learning model (Blanken et al. 2022), the prior use of Gan in ULM to obtain good performance in vivo at high concentration (Gu et al. 2023). In addition, it does not cover several prior works using the temporal context in the architecture of the network (Chen et al. 2022; Milecki et al. 2021, Xing et al., IEEE TMI, 2023). The manuscript would benefit from a more focused introduction on the problem addressed by the manuscript (i.e., producing better training data and improving the deep learning model).

Response Thank you for the valuable suggestions. We have integrated the referenced works in the Introduction to broaden the scope of our literature review.

(Blanken et al. 2022) **Page 4 Line 90** To account for these effects, a simulation framework was developed to model nonlinear wave propagation and MB responses, generating single-channel RF signals obtained from an MB cloud¹⁹. A 1D CNN was then trained to localize and deconvolve MB signals from raw RF data. This approach achieved a tenfold increase in the axial resolution of beamformed images using deconvolved RF data compared to standard B-mode images. However, this technique requires precise parameter tuning and system characterization to generate realistic datasets for training.

(Gu et al. 2023, Milecki et al, 2021) **Page 5 Line 120** In contrast, ULM-GAN²² circumvents the conventional localization-and-tracking approach by using a training strategy that directly maps the temporal average of short-accumulation ultrasound images to ULM images accumulated from long data acquisitions. Additionally, the use of spatio-temporal 3D-convolutional neural networks (3D-CNN) enables the direct extraction of dense MB networks at high MB concentrations²³. However, these techniques have thus far been limited to spatial reconstructions of microvasculature, which omits potentially critical physiological biomarkers such as blood flow velocity.

(Chen et al. 2022; Milecki et al. 2021, Xing et al., IEEE TMI, 2023) **Page 5 Line 130** Our work also expands on the application of spatio-temporal networks, such as 3D-CNNs and Long Short-Term Memory networks in super-resolution imaging. These networks have been successfully utilized in ultrasound MB imaging for tasks like phase aberration correction¹⁶, localization-free microvessel velocimetry²⁵, and MB track reconstruction²³. The DECODE network aims to enhance MB localization performance in high-density MB regime by integrating spatiotemporal information across adjacent frames.

R2.5) The computation performance subsection in the results is not convincing as it only relies on the time measurement of processing with fixed parameters. In addition, it is not clear if the LOCA-ULM and Conventional ULM computations are made on GPU or CPU. In the case where they are not performed on the same device, it should be specified as a limitation of the comparison and both CPU times could also be provided to provide some perspective.

Response In response to similar concerns raised by Reviewer 3 Comment 7 (R3.7), the section on computational performance has been updated to report objective comparisons of computational times. The complexity of the conventional algorithm (normalized cross-correlation implemented by MATLAB's `normxcorr2`) can be approximated as $O(MN \log(MN))$ for an image of size $M \times N$. Additionally, the `imregionalmax` function from MATLAB was used for MB localization, which has a complexity of $O(MN)$.

However, quantifying the computational complexity of deep learning architectures like mSPCN, Deep-ULM, and LOCA-ULM is less straightforward due to multiple influencing factors. These include network depth, number of kernels per convolutional layer, convolutional operations such as stride and padding, and input image size. Consequently, a direct comparison using the Big O notation was not feasible. As a result, we conducted an experimental assessment of each method's computational performance using a consistent GPU setup based on an NVIDIA RTX A6000 GPU. Computational performance of four localization algorithms (conventional ULM, Deep-ULM, mSPCN, and LOCA-ULM) was evaluated using 100 simulated imaging frames with increasing sizes of field-of-view (FOV). The computation time is presented in Fig. 2f (also presented below), which includes network forward processing and the subsequent post-processing for localization. LOCA-ULM processed 100 frames with a 139.88 mm^2 FOV in approximately 1.7 seconds, which was 5.6 times faster than the conventional method. Furthermore, LOCA-ULM directly estimates the MB centers, bypassing the process of peak identification required by conventional ULM, Deep-ULM, and mSPCN. This efficiency translates to a two-fold acceleration for LOCA-ULM over

mSPCN (3.6 seconds) and Deep-ULM (3.5 seconds) when processing 100 frames with a 139.88 mm^2 FOV. The new results have been added to the Results and the Fig. 2f.

Fig 2. f Comparison of computational times for MB localization across varying FOV sizes using conventional ULM, Deep-ULM, mSPCN and LOCA-ULM. Computational times encompass the MB localization process, which includes normalized cross-correlation and regional maximum search for conventional ULM, a network forward pass for LOCA-ULM, and a network forward pass with centroid calculation for both Deep-ULM and mSPCN.

Page 11 Line 258 *LOCA-ULM significantly improves computational performance of MB localization*

Computational performance of four localization algorithms (conventional ULM, Deep-ULM, mSPCN, and LOCA-ULM) was evaluated using 100 simulated imaging frames with different sizes of field-of-view (FOV). The FOV ranged from 0.40 mm^2 ($16 \text{ pixel} \times 16 \text{ pixel}$ area) to 139.88 mm^2 ($300 \text{ pixel} \times 300 \text{ pixel}$ area), with a 4-pixel increment step (in each dimension). The pixel resolution was fixed at $39.4 \mu\text{m}$. For each FOV size, the number of MBs simulated in each frame also increased from 16 to 300 in increments of 4. Fig. 2f presents the computational time results for the four different localization methods. Computational times encompass the MB localization process, which includes normalized cross-correlation and regional maximum search for conventional ULM, a network forward pass for LOCA-ULM, and a network forward pass with centroid calculation for both Deep-ULM and mSPCN. Conventional localization based on normalized cross-correlation exhibits a steep increase in computational time as the FOV expands (i.e., number of pixels increases). In contrast, all deep learning-based methods demonstrate improved computational performance with LOCA-ULM consistently showing the lowest computation time across all FOVs. At an FOV of 139.88 mm^2 , the computational time for LOCA-ULM was approximately 1.7 seconds, representing a 5.6-fold acceleration compared to the conventional localization (9.5 seconds). Moreover, LOCA-ULM directly estimates the MB centers, therefore bypassing the process of peak identification required by Deep-ULM and mSPCN. This efficiency translates to LOCA-ULM achieving a two-fold acceleration compared to mSPCN (3.6 seconds) and Deep-ULM (3.5 seconds) for an FOV of 139.88 mm^2 . All methods were evaluated on an NVIDIA RTX A6000 GPU.

R2.6) The authors need to better describe the tracking process along with a discussion on the potential biases it may cause. For instance, since LOCA-ULM is based on spatiotemporal datasets, it would be reasonable to expect longer track lengths. One could imagine obtaining longer track lengths simply

by temporally filtering datasets in conventional ULM, which may lead to an improved connection between vessels.

Response We thank the reviewers for their constructive comments. First, we have added a detailed description of the uTrack algorithm used in our study in the Method section:

Page 29 Line 708 The centroid coordinates obtained by different localization methods were input into the uTrack algorithm⁴⁰. uTrack solves tracking as a two-step linear assignment problem. The first step involves establishing frame-to-frame MB linking to generate initial track segments. In the second step, uTrack connects these track segments across the entire time-lapse sequence to form complete trajectories. The MB tracks were generated with a minimum persistence of 10 frames. Furthermore, links between track segments with a linking angle exceeding 45° were considered unreliable and rejected.

We appreciate the reviewer's interest in the potential biases introduced by the tracking process. We agree that the spatio-temporal CNN architecture of LOCA-ULM could result in longer MB track lengths. However, these extended MB tracks might also be attributed to the improved localization accuracy for overlapping and intersecting MBs, which reduces the rate of missed MBs within tracks. Quantifying the impact of spatio-temporal network and enhanced localization performance of LOCA-ULM on tracking performance requires a dedicated future study for an in-depth analysis.

While examining the impact of the LOCA-ULM on MB tracks is beyond the scope of the current study, we invite attention to the localization results within our paper. These results demonstrate that LOCA-ULM outperforms both conventional and other deep learning methods (i.e., Deep-ULM and mSPCN) in MB localization without tracking (refer to Fig. 2 a,b, Fig. 4 i,j). We hypothesize that the enhanced localization performance of LOCA-ULM facilitates better tracking, manifested by an increased number of MB tracks and their connectivity as shown in Fig. 3, 4, and 5. Nevertheless, we acknowledge this hypothesis requires validation through future studies. At present, the LOCA-ULM pipeline itself does not perform tracking; therefore, the speculated benefits on tracking performance (e.g., track lengths) are yet to be confirmed.

Specific comments

R2.7) Line 20: 'revolutionized' is too strong.

Response We have modified 'revolutionized' to 'advanced.'

Page 1 Line 20 Ultrasound localization microscopy (ULM) has **advanced** deep tissue microvascular imaging by enabling the reconstruction of vasculature at a microscopic scale through the localization and tracking of intravenously injected microbubbles (MBs).

R2.8) Line 25: missing word

Response We have modified the statement to:

Page 1 Line 23 Here, we introduce Localization with Context Awareness (LOCA)-ULM, a deep learning-based MB simulation and localization pipeline designed to enhance MB localization performance in high MB densities.

R2.9) Line 52: Claim is too strong – while it is true that resolution is better maintained with depth in ULM, a larger PSF leads, typically, to a poorer resolution in ULM.

Response We have modified the main text to:

Page 3 Line 54 When combined with deep penetration of ultrasonic waves, the accurate localization of MBs offers the potential for reconstructing the deep vascular network with micron-scale spatial resolution.

R2.10) Line 83: SIMUS (Garcia, Comput Methods Programs Biomed, 2022) has also been used in deep-learning methods for ULM.

Response We have added reference “Xing et al., IEEE TMI, 2023” and cited SIMUS accordingly.

Page 4 Line 82 Existing ultrasound modeling and simulation software, such as bivariate Gaussian models¹⁴, Field II simulations¹⁵, and SIMUS¹⁶ have been used to simulate MB signals.

R2.11) Line 154: please clarify what ‘real ultrasound data’ means.

Response We have modified ‘real ultrasound data’ to ‘*in vivo* contrast-enhanced ultrasound images’.

Page 7 Line 177 In the inference stage (Fig. 1e), the network estimates MB locations and brightness levels with *in vivo* contrast-enhanced ultrasound images as input.

R2.12) Line 203: minimal resolution, not maximal

Response We appreciate the opportunity to clarify the terminology used. In the context of our study, ‘maximal resolution’ refers to the highest achievable resolution, which is a term frequently utilized in the field, as exemplified by the works of Desailly et al., Physics in Medicine & Biology (2015) and Foiret et al., Scientific Reports (2017). These references use the term maximal resolution to describe the highest resolution at which single and isolated ultrasound source can be localized. We have revised “maximal resolution” to “highest resolution” to maintain consistency with established literature.

Page 9 Line 223 The theoretical resolution limit of ULM (i.e., localization error) can be estimated using the Cramér-Rao lower bound (CRLB)²⁹, which gives a **highest** resolution of 3.29 μm with the CAM study acquisition settings (Methods).

R2.13) Line 278: It is unclear that the value in parenthesis (49.28 μm) refers to the half wavelength and not to the wavelength.

Response The modification now explicitly mentions that 49.28 μm is the half wavelength.

Page 13 Line 325 Our results showed that both LOCA-ULM and conventional localization produced a spatial resolution that is below a half wavelength at the imaging frequency of 15.625 MHz (that is, 49.28 μm) regardless of the application of MB separation (Supplementary Fig. 2).

R2.14) Line 282: The sentence needs to be clarified.

In response to the similar concerns raised by Reviewer 3 Comment 16 (**R3.16**), we agree that the statement is difficult to verify and therefore revised the manuscript to remove the following statement.

“Notably, LOCA-ULM reconstructed a greater number of vessels without degrading the spatial resolution as indicated by the FRC. While strategies such as relaxed MB intensity or cross-correlation cutoff could lead to increased localizations when using conventional localization techniques, they could also lead to false localizations which undermine spatial resolution.”

R2.15) Line 303: Please define what is meant by ‘tiny’ or remove.

Response We have removed ‘tiny.’

R2.16) Line 331: The authors could consider broadening their claims to dynamic ULM methods in general and not only fULM.

Response Thank you for your suggestion. We agree that LOCA-ULM extends beyond fULM and accordingly expanded our Discussion to encompass dynamic:

Page 21 Line 517 Furthermore, LOCA-ULM could potentially improve dynamic ULM (DULM)³³ by increasing the number of localized and tracked MBs, offering more velocity measurements over time and potentially decreasing the total acquisition time by relaxing the need for data acquisitions from multiple cardiac cycles.

R2.17) Line 496: Please clarify the model used for the random fluctuations.

Response We have clarified the random fluctuations used in our MB motion model.

Page 23 Line 575 To simulate stochastic MB motion, the direction of the MBs was perturbed at every time step by adding a small random vector. The components of this random vector were sampled from a normal distribution with a standard deviation of 0.2. After the random direction perturbation, the direction vector and the speed are multiplied to obtain the velocity. The MB position was updated according to the velocity, and the position at each time frame was stored as the ground truth.

Reviewer #3 (Remarks to the Author):

Please note that the Point Q7 was not fully shown in the file. Please see it below:

R3.1) General comment on superlatives and potentially exaggerated claims. The article uses too many superlatives, which reduces the overall quality of the article as we cannot distinguish a real improvement from an exaggerated claim. Please be as quantitative as possible. Revise the abstract and the end of the discussion. See also questions Q6 and Q7.

Response Thank you for your feedback. We have carefully revised the manuscript and removed superlatives and exaggerated claims, focusing instead on presenting our findings with quantitative data. These modifications have been applied consistently across the entire manuscript. We believe these revisions improved the clarity of objectivity of the manuscript.

Page 1 Line 20 Abstract

Ultrasound localization microscopy (ULM) has advanced deep tissue microvascular imaging by enabling the reconstruction of vasculature at a microscopic scale through the localization and tracking of intravenously injected microbubbles (MBs). However, the localization process in current ULM is limited by the requirement for spatially isolated MBs. Here, we introduce Localization with Context Awareness (LOCA)-ULM, a deep learning-based MB simulation and localization pipeline designed to enhance MB localization performance in high MB densities. *In silico* experiments showed that LOCA-ULM achieved an average MB detection accuracy of 97.7% and reduced the missing rate to 24.5%, outperforming the average detection accuracy and missing rate of Deep-ULM (88.2%, 51.4%), mSPCN (89.4%, 61.6%), and conventional localization (80.3%, 62.7%). In *in vivo* rat brain imaging studies, LOCA-ULM achieved higher MB localization rate and accuracy compared to conventional localization across all tested MB concentrations, enabling the reconstruction of dense vascular networks as well as the identification of adjacent small and large vessels that were missed by conventional ULM. In the functional ULM (fULM) imaging experiment, LOCA-ULM revealed activations within small vessels that were undetectable by conventional ULM, achieving a two-fold increase in MB count and 1.85-fold increase in activated pixel area within the somatosensory and thalamus regions associated with whisker stimulation. The increase in MB concentration and reduction in acquisition time makes LOCA-ULM a highly efficient and practical tool for future applications of ULM in the wide range of preclinical and clinical applications that involve tissue microvasculature as a biomarker.

R3.2) Figure 2.c and 2.d. The LOCA-ULM is compared with the Conventional ULM (no MB separation). This is not the best because the spectrum separation (Conventional MB separation) is better and has practically the same computational cost. Spectrum separation is a trivial procedure that enables to increase the number of detected bubbles you must always compare with a MB separation (see also Q6).

Response Thank you for your feedback regarding the comparison between LOCA-ULM and conventional ULM methods in the context of MB separation. In light of the reviewer's comments, we have extensively compared LOCA-ULM against conventional ULM with MB separation in a rat brain study and expanded our experiments. The detailed response is provided in Comment 6 below.

Moreover, our initial comparison analysis for CAM imaging was designed to evaluate the baseline performance of LOCA-ULM against conventional ULM without MB separation. While MB separation is a simple process that could enhance the detection rate of MBs, it also substantially increases the processing time by splitting the original data into multiple subsets. Each subset then undergoes separate localization and tracking process. Therefore, we believe it is beneficial to compare the performance of different localization algorithms without MB separation.

R3.3) Figure 2.d In the image I estimate that the vessels have a minimum separation of $\sim 0.07\text{mm}$. At a relatively high frequency of 20MHz the wavelength is only 0.07mm, so this image does not show super resolution in the sense that we can separate two vessels below the diffraction limit. Of course, the vessels are highly contrasted, and the diameters are very similar to the ground truth. Can you comment on that? Why did you use 20 MHz in this experiment and 15 MHz in the rat?

Response This is a good point. We would like to clarify that the “missing” small vessels in the CAM ULM images are not caused by limited imaging resolution of ULM, but rather due to anatomical factors: the smaller vessels or capillaries of the CAM are located at the superficial layer of the membrane, which is in direct contact with air (Chen, L et al., *Cells* (2021)). They are thus not accessible to ultrasound imaging because of the strong acoustic impedance mismatch between air and soft tissue. This can be confirmed with ultrasound imaging, shown in Fig. R3 below, where MBs appear to “vanish” from the ultrasound imaging FOV when traversing the vascular bed between an artery and a vein. Another reason that could explain the difficulties of imaging capillaries in the CAM is the low compliance of these small vessels which could restrict the oscillations of the MB shell, leading to reduced MB backscattering signal (Lowerison, M. R., et al., *IEEE TUFFC* (2020)). Despite these challenges, the CAM imaging setup remains the best *in vivo* validation platform for ULM due to the availability of an optical imaging reference. Measurements such as vessel size comparisons still provide meaningful results for the purpose of validating ULM reconstruction.

Fig. R3 Ultrasound MB signal (right) and corresponding optical image (left) acquired from a CAM microvessel model. Reprinted from *In Vivo Confocal Imaging of Fluorescently Labeled Microbubbles: Implications for Ultrasound Localization Microscopy*, by M. R. Lowerison, et. al., in *IEEE Transactions on Ultrasonics, Ferroelectrics, and Frequency Control*, vol. 67, no. 9, pp. 1811-1819, 2020, Reprinted with permission.

Regarding the choice of transducer frequencies, we selected the L35 transducer for the CAM experiments to achieve the best possible imaging resolution. Although the nominal center frequency of the L35 is 25 MHz, our observations indicated a higher sensitivity at 20 MHz. Therefore, we chose to transmit the probe at 20 MHz center frequency instead of 25 MHz. For the rat brain experiments, we used a 15 MHz L22 transducer to leverage its larger FOV and deeper penetration, which are better suited for the anatomical requirements of rat brain imaging.

- Chen, Lei, et al. "Utilisation of chick embryo chorioallantoic membrane as a model platform for imaging-navigated biomedical research." *Cells* 10.2 (2021): 463
- Lowerison, Matthew R., et al. "In vivo confocal imaging of fluorescently labeled microbubbles: Implications for ultrasound localization microscopy." *IEEE transactions on ultrasonics, ferroelectrics, and frequency control* 67.9 (2020): 1811-1819

R3.4) Line 171. LOCA-ULMe and LOCA-ULM are confusing. Can you clarify the methods? As it stands, I cannot see the interest in introducing the LOCA-ULMe, which is only proposed in the chicken embryo experiment.

Response Thank you for pointing this out. First, to enhance the clarity of the two methods, we have revised the terminology from LOCA-ULM^E to LOCA-ULM^{Experimental} and LOCA-ULM to LOCA-ULM^{LSGAN}. LOCA-ULM^{Experimental} was a preliminary training approach in which the LOCA-ULM network was trained using MB signals directly extracted from *in vivo* CAM data, which is essential for establishing a baseline performance. In contrast, standard LOCA-ULM^{LSGAN} was trained on synthetic MB templates generated by the LSGAN network, which itself was trained using the same set of *in vivo* MB signals as LOCA-ULM^{Experimental}. We conducted experiments with both LOCA-ULM^{LSGAN} and LOCA-ULM^{Experimental} to examine the benefit of having augmented MB datasets generated by the LSGAN network. In this case, LOCA-ULM^{Experimental} served as the baseline for the comparison study and the results showed that the MB data augmentation by LSGAN was indeed beneficial for improving MB localization performance (Fig. 2d, e). We have revised the manuscript to clarify this study design.

Page 10 Line 233 To demonstrate the effectiveness of LSGAN-generated MBs, we trained the LOCA-ULM network using two distinct simulation datasets. The first, referred to as LOCA-ULM^{Experimental} was trained using MB signals directly extracted from the *in vivo* CAM data. The second, LOCA-ULM^{LSGAN}, was trained with synthetic MB templates generated by the LSGAN network trained on the same set of MB signals used to train the LOCA-ULM^{Experimental}

Page 10 Line 253 The findings suggest that synthetic MB signals generated by LSGAN act as a form of data-augmentation, allowing LOCA-ULM^{LSGAN} to learn from a broader distribution of MB signals—specifically, MB signals that were not part of LOCA-ULM^{Experimental} training dataset, and improve the localization performance of the DECODE network.

R3.5) line 247- 283. In this experiment the concentration is not explicit (main text or mat&met). This is important because you show later (Figure 4.j) that there is no benefit from using LOCA-ULM at low concentrations. I assume that the concentration is low (20ul/min?).

Response In the rat brain study depicted in Fig. 3, we employed an injection rate of 15 $\mu\text{L}/\text{min}$ and the description has been added in the revised manuscript. Moreover, Fig. 4j illustrates that LOCA-ULM achieved an approximately 1.5-fold increase in the number of MBs localized with MB separation at a low concentration (20 $\mu\text{L}/\text{min}$). This result leads to enhanced visualization of the vasculature, as evidenced by the enhanced delineation of adjacent vessels and reconstruction of connected vessel networks (red and green arrows in Fig. 4 e,f). Detailed evidence of the superior performance of LOCA-ULM across a range of concentrations (ranging from 20 $\mu\text{L}/\text{min}$ to 40 $\mu\text{L}/\text{min}$ in Fig.4 and 90 $\mu\text{L}/\text{min}$ in Supplementary Fig. 4), is detailed in our response to Comment 6 below.

Page 12 Line 297 Fig. 3c, d shows the final ULM images based on 20000 frames (a total of 80 seconds of data acquisition) of accumulation with an MB injection rate of 15 $\mu\text{L}/\text{min}$ (Methods).

Page 31 Line 760 In the rat brain study (illustrated in Fig. 3), we used an injection rate of 15 $\mu\text{L}/\text{min}$. For the study of comparing the performance of different localization methods in different MB concentrations (illustrated in Fig. 4), we varied the injection rate to 20, 30, 40 $\mu\text{L}/\text{min}$ with a 3-minute waiting period after changing the injection rate to stabilize the systemic MB concentration.

R3.6) line 247- 283. The title of the section is "LOCA-ULM demonstrates superior in vivo ULM imaging performance in a rat brain", but at low concentrations both methods (LOCA-ULM and ULM with MB separation) are relatively equivalent. It is only at high concentrations that LOCA-ULM becomes interesting. If we compare the LOCA-ULM with the conventional ULM (Fig. 3.c and d), we observe a clear gain in the LOCA-ULM, but if we compare it with the conventional WITH directional separation, this gain is relatively moderate. It is important to note that the directional separation is very simple (we split the spectrum in 2). For this reason, the comparison with the standard ULM without separation is of no interest, we must focus on the comparison with the state of the art, which is WITH separation.

Response Thank you for your insightful comment. We designed the original experiments to utilize an MB concentration corresponding to an MB injection rate of $15 \mu\text{L} / \text{min}$, as illustrated in Fig. 3. This is where conventional localization begins to fail without MB separation. To the best of our knowledge, the general ULM community still uses standard MB localization without MB separation. Our intention was to use the most basic MB localization technique (i.e., without MB separation) as the baseline for comparing localization performances. Furthermore, as discussed in our 2019 paper that introduced MB separation (Huang, C. and Lowerison, M. R., et al., Sci Rep (2019)), we would like to emphasize that while the concept of MB separation is straightforward, it significantly increases the computational load.

We agree with the reviewer that the improvement of LOCA-ULM over conventional localization with MB separation was not as significant as without MB separation at the lowest MB concentration ($15 \mu\text{L} / \text{min}$, as shown in Fig. 3). However, we observed an improvement at higher MB concentrations ranging from $20 \mu\text{L} / \text{min}$ to $40 \mu\text{L} / \text{min}$. LOCA-ULM enhanced delineation of adjacent vessels (green arrows in Fig. 4 e,f,g,h) for both the $20 \mu\text{L} / \text{min}$ and $40 \mu\text{L} / \text{min}$ injection rates. Moreover, LOCA-ULM improved the visualization of the vascular network by more accurately reconstructing the connected vessel structures branching from the main vessels (red arrows in Fig. 4e,f,g,h). This improvement is supported by the saturation of the number of localized MBs for conventional localization with MB separation at a low injection rate ($20 \mu\text{L} / \text{min}$ in Fig. 4j). In contrast, LOCA-ULM demonstrates superior localization performance, localizing approximately 1.5 times more MBs than conventional localization, with MB separation. LOCA-ULM also demonstrates a progressive increase in number of localized MBs as the concentration increases, indicating it is not saturated by high MB concentrations.

Finally, we have added a new experiment where a very high MB concentration ($90 \mu\text{L} / \text{min}$) was used to further characterize the gain in MB localization performance for LOCA-ULM. As shown in Supplementary Fig. 4, the advantage of LOCA-ULM over conventional localization with MB separation is clear: LOCA-ULM was able to recover many more vessels that were missing from the conventional ULM with MB separation. While our current LOCA-ULM framework demonstrates significant improvement without the need for MB separation (Fig. 3 and Fig. 4), we agree that MB separation could be implemented as it further improves the vessel delineation, particularly in very high MB concentration (Supplementary Fig. 4). The associated increase in computational load using MB separation should be carefully considered against the expected improvements in image quality and localization precision. These new results, along with new discussions, have been added to the revised manuscript as follows:

Page 15 Line 360 When comparing LOCA-ULM with the conventional ULM with state-of-the-art MB separation, LOCA-ULM provides improved separation of adjacent vessels with enhanced contrast in both $20 \mu\text{L} / \text{min}$ and $40 \mu\text{L} / \text{min}$ injection rates (green arrows in Fig. 4 e-h). LOCA-ULM also improved the visualization of the vascular networks by more accurately reconstructing the connected vessel structures branching from the main vessels (red arrow in Fig. 4 e-h). Further comparison of LOCA-ULM with MB separation at $90 \mu\text{L} / \text{min}$ injection rate (Supplementary Fig. 4) shows that LOCA-ULM achieves robust reconstruction even in challenging conditions where conventional ULM with MB separation fails.

Page 15 Line 373 With the addition of MB separation (Fig. 4j), conventional localization showed improved localization efficacy. However, the number of MB localized plateaued at a low injection rate of $20 \mu\text{L}/\text{min}$. In contrast, LOCA-ULM demonstrated superior localization performance, localizing approximately 1.5 times more MBs at low concentration ($20 \mu\text{L}/\text{min}$) compared to conventional localization, and showing a progressive increase in number of localized MBs as the concentration increased.

- Huang, Chengwu, et al. "Short acquisition time super-resolution ultrasound microvessel imaging via microbubble separation." *Scientific reports* 10.1 (2020): 6007

Fig 4. Effect of different MB injection rates ($20 \mu\text{L}/\text{min}$, $30 \mu\text{L}/\text{min}$, $40 \mu\text{L}/\text{min}$) on LOCA-ULM and conventional localization for rat brain ULM imaging. e-h, with MB separation. i,j Comparison of total MB count per acquisition (a total of 250 frames per acquisition) for LOCA-ULM and conventional localization at

different MB injection rates ($20 \mu\text{L}/\text{min}$, $30 \mu\text{L}/\text{min}$, and $40 \mu\text{L}/\text{min}$). Two datasets, **i** without MB separation and **j** with MB separation.

Supplementary Fig 4. Comparison of conventional ULM and LOCA-ULM (with and without MB separation) at a very high MB concentration. Each ULM image was generated by accumulating 16000 frames of ultrasound data (20 seconds) for an MB injection rate of $90 \mu\text{L}/\text{min}$.

R3.7) Line 234-245. Computational performance. Avoid repetitive superlatives about your method, and focus exclusively on computational time, not on repeating previously discussed advantages. The reader needs to understand for themselves that it is better. For example, "In addition to faster and more robust vessel filling performance" or « Collectively, these results indicate that not only does LOCA-ULM provide more robust MB localization » There is no doubt that the LOCA-ULM can provide a faster algorithm, but the discussion about computation time is too simplistic. Computing time is highly dependent on the implementation of the algorithms. Please rewrite this section to include more useful information. What is the order of complexity of the algorithms (for example is

an algorithm $O(2)$ or $O(3)$?), can you compare the total number of operations in both algorithms? Differences in the order of complexity of the algorithms or in the total number of operations can provide useful information about why your algorithm is better. Are both algorithms implemented in GPU? please include this information in the main text.

Response In response to similar concerns raised by Reviewer 2 Comment 5 (R2.5), We have revised computational performance section to reflect objective comparison of computational time without the use of superlatives. The complexity of the conventional algorithm (normalized cross-correlation implemented by MATLAB's `normxcorr2`) can be approximated as $O(MN\log(MN))$ for an image of size $M \times N$. Additionally, the `imregionalmax` function from MATLAB was used for MB localization, which has a complexity of $O(MN)$.

However, quantifying the computational complexity of deep learning architectures like mSPCN, Deep-ULM, and LOCA-ULM is less straightforward due to multiple influencing factors. These include network depth, number of kernels per convolutional layer, convolutional operations such as stride and padding, and input image size. Consequently, a direct comparison using the Big O notation was not feasible. As a result, we conducted an experimental assessment of each method's computational performance using a consistent GPU setup based on an NVIDIA RTX A6000 GPU. Computational performance of four localization algorithms (conventional ULM, Deep-ULM, mSPCN, and LOCA-ULM) was evaluated using 100 simulated imaging frames with increasing sizes of field-of-view (FOV). The computation time is presented in Fig. 2f (also presented below), which includes network forward processing and the subsequent post-processing for localization. LOCA-ULM processed 100 frames with a 139.88 mm^2 FOV in approximately 1.7 seconds, which was 5.6 times faster than the conventional method. Furthermore, LOCA-ULM directly estimates the MB centers, bypassing the process of peak identification required by conventional ULM, Deep-ULM, and mSPCN. This efficiency translates to a two-fold acceleration for LOCA-ULM over mSPCN (3.6 seconds) and Deep-ULM (3.5 seconds) when processing 100 frames with a 139.88 mm^2 FOV. The new results have been added to the Results and the Fig. 2f.

Fig 2. f Comparison of computational times for MB localization across varying FOV sizes using conventional ULM, Deep-ULM, mSPCN and LOCA-ULM. Computational times encompass the MB localization process, which includes normalized cross-correlation and regional maximum search for conventional ULM, a network forward pass for LOCA-ULM, and a network forward pass with centroid calculation for both Deep-ULM and mSPCN.

Computational performance of four localization algorithms (conventional ULM, Deep-ULM, mSPCN, and LOCA-ULM) was evaluated using 100 simulated imaging frames with different sizes of field-of-view (FOV). The FOV ranged from 0.40 mm^2 (16 pixel \times 16 pixel area) to 139.88 mm^2 (300 pixel \times 300 pixel area), with a 4-pixel increment step (in each dimension). The pixel resolution was fixed at $39.4 \mu\text{m}$. For each FOV size, the number of MBs simulated in each frame also increased from 16 to 300 in increments of 4. Fig. 2f presents the computational time results for the four different localization methods. Computational times encompass the MB localization process, which includes normalized cross-correlation and regional maximum search for conventional ULM, a network forward pass for LOCA-ULM, and a network forward pass with centroid calculation for both Deep-ULM and mSPCN. Conventional localization based on normalized cross-correlation exhibits a steep increase in computational time as the FOV expands (i.e., number of pixels increases). In contrast, all deep learning-based methods demonstrate improved computational performance with LOCA-ULM consistently showing the lowest computation time across all FOVs. At an FOV of 139.88 mm^2 , the computational time for LOCA-ULM was approximately 1.7 seconds, representing a 5.6-fold acceleration compared to the conventional localization (9.5 seconds). Moreover, LOCA-ULM directly estimates the MB centers, therefore bypassing the process of peak identification required by Deep-ULM and mSPCN. This efficiency translates to LOCA-ULM achieving a two-fold acceleration compared to mSPCN (3.6 seconds) and Deep-ULM (3.5 seconds) for an FOV of 139.88 mm^2 . All methods were evaluated on an NVIDIA RTX A6000 GPU.

R3.8) Figures 4 and 5. The colors in Figures 4 and 5 are different from those in Figure 3. This makes it very difficult to compare the figures. Please use a single-color code for both figures, it is also appreciated to include a detail of the cortical part in all cases as in figure 3.

Response Thank you for the suggestions. We have modified Figures 4 and 5 to have the identical colormap with Figure 3 (copied below for the reviewer's convenience). We have also added zoomed ROI of the cortical regions for all cases.

R3.9) Figure 4. Why do you stop the concentration at 40ul/min in this experiment? In the fUS experiment the concentration is increased to 60ul/min. Do you reach saturation of the LOCA-ULM at 60ul/min?

Response Regarding Figure 4 and the selected concentration threshold of 40 $\mu\text{L} / \text{min}$, the experimental design for the rat brain imaging and the fULM imaging were decided on distinct methodological requirements. Our results demonstrates that the localization efficiency of conventional ULM plateaus at an MB injection rate of 20 $\mu\text{L} / \text{min}$, with or without MB separation (illustrated in Fig. 4i, j). Therefore, we determined that the concentration of 40 $\mu\text{L} / \text{min}$ was sufficient to demonstrate superior performance of LOCA-ULM against conventional ULM, without necessitating higher MB concentrations.

The fULM experiment required a higher MB concentration (60 $\mu\text{L} / \text{min}$) due to the temporal sparsity of the MB signal, which is a challenge specific to fULM imaging. We did not extend our analysis to 60 $\mu\text{L} / \text{min}$ in the experiments presented in Fig. 4, and we acknowledge that at such high concentrations, LOCA-ULM could reach a saturation point. The MB concentration for fULM experiment was selected based on the intrinsic trade-off: higher MB concentrations are required to overcome temporal MB signal sparsity, yet this increase may compromise the optimal quality of ULM images. Consequently, we increased the MB concentration to 60 $\mu\text{L} / \text{min}$ to enhance the sensitivity and eliminate sparse signals. Nevertheless, it is shown in the added Supplementary Fig. 5 (copied below) that LOCA-ULM can still produce superior ULM reconstructions under high MB concentration (60 $\mu\text{L} / \text{min}$), showing enhanced contrast and finer vascular details compared to conventional ULM (white arrows in Supplementary Fig. 5).

Fig 4. i,j Comparison of total MB count per acquisition (a total of 250 frames per acquisition) for LOCA-ULM and conventional localization at different MB injection rates (20 $\mu\text{L}/\text{min}$, 30 $\mu\text{L}/\text{min}$, and 40 $\mu\text{L}/\text{min}$). Two datasets, **i** without MB separation and **j** with MB separation.

Supplementary Fig 5. Comparison of conventional ULM and LOCA-ULM for functional ULM (fULM) experiment. Each ULM images were generated by accumulating 720000 frames of ultrasound data for MB injection rate of 60 $\mu\text{L}/\text{min}$.

R3.10) Lines 320-330 Figure 6. Two other methods are presented here (Deep-ULM and mSPCN) but there is no bibliography, and they are not mentioned in the methods section. Include the concentration in the main text (the 40ul/min is only in the caption).

Response Thank you for your comment. We have provided bibliography of Deep-ULM and mSPCN in both Figure 2 and 5 and in the Introduction where the two models are first mentioned:

Fig 2. Results of the simulation study and in vivo chicken embryo CAM imaging study. a Simulation results of conventional localization, Deep-ULM¹⁴, mSPCN²¹, and LOCA-ULM with low (0.16 MBs/λ^2) and high (0.35 MBs/λ^2) MB concentrations.

Fig 5. Comparison among LOCA-ULM, conventional ULM, Deep-ULM¹⁴, mSPCN²¹ in in vivo rat brain ultrasound data under high MB concentration (injection rate 40 $\mu\text{L}/\text{min}$)

Page 5 Line 116 Approaches such as Deep-ULM¹⁴ and modified sub-pixel convolutional neural network (mSPCN)²¹ employ mean-squared error-based regression to transform input contrast-enhanced ultrasound images to super-resolved images.

Moreover, we have expanded the Supplementary Methods section to include detailed descriptions of the implementation of Deep-ULM and mSPCN.

Supplementary Methods 3 This study employed two additional deep learning-based ULM methods to compare with LOCA-ULM: Deep-ULM¹⁴ and mSPCN²¹. Deep-ULM used the U-Net architecture where the encoder consists of three stages, each containing two convolutional layers with 3×3 kernels, batch normalization, leaky ReLU activation, and a 2×2 MaxPooling operation. The latent layer, positioned between the encoder and the decoder, includes two convolutional layers with 3×3 kernels and a dropout layer (probability 50%). The decoder consists of three stages with deconvolution layers with 5×5 kernels (first layer with a stride of 2 and the second a stride of 1), batch normalization, and leaky ReLU activation. The first two stages of the decoder also incorporate a 2×2 upsampling layer. The final layer is a single-channel output convolutional layer with linear activation. The mSPCN model (provided by the authors in DOI: 10.21227/jdgd-0379) employs a sub-pixel architecture that consists of 13 convolution layers. It begins with a 9×9 kernel convolution layer for feature extraction, followed by ten 3×3 convolution kernel convolution layers. Residual blocks are integrated every two layers. The 12th convolution layer forms a global residual connection to the first layer, while the final layer performs the upscaling operation using a sub-pixel convolution layer. We trained both networks using the Adam optimizer with a learning rate of 0.001 for 400 epochs, aiming to minimize the following cost function:

$$\mathcal{L}(x, y|\theta) = \|f(x|\theta) - \lambda_0 y\|_2^2 + \lambda_1 \|f(x|\theta)\|_1$$

where x denotes the simulated ultrasound MB image, and y represents the super-resolved image. The coefficients were set as $\lambda_0 = 100$ and $\lambda_1 = 0.1$ for both networks. The mSPCN and Deep-ULM models operate as image-to-image translation networks, where the output depicts the center of MB as a cluster of activated pixels rather than a single pixel. In line with the method outlined in mSPCN²⁷, the MB positions were determined by applying a 2D Gaussian kernel to group the adjacent non-zero pixels, and the centroid is identified by computing the center of mass.

The reviewer was correct that the MB injection rate was 40 $\mu\text{L}/\text{min}$, which has been added to the Results section of the main text.

Page 16 Line 383 In Fig. 5a, we compared the MB localization performance of LOCA-ULM with two other deep learning-based MB localization methods, Deep-ULM and mSPCN, on the high MB concentration rat brain dataset (40 $\mu\text{L}/\text{min}$) from the previous section.

R3.11) Lines 331-378. The functional experiment is an interesting result. Can you compare it to a standard correlation with power Doppler (if you did the experiment)? In my opinion, both methods are complementary, standard FUS could be more sensitive and identify the regions without ambiguity. Outside the activated regions, the LOCA-ULM image has a lot of negative correlations and also mixed positive and negative correlations that are difficult to analyse. Are these significant or random noise? You can add a supplementary figure showing only the significant activated pixels.

Please also explicitly mention in the main text that the concentration (60ul/min) was higher than the previous ones.

Response Thank you for the constructive suggestions. Below please find our detailed response to your question:

1. Comparison with standard fUS based on power Doppler: We have now included functional ultrasound (fUS) imaging results obtained through ultrafast Doppler imaging (contrast-free) in Fig. 6c (Macé, E., Montaldo, G., Cohen, I. et al., *Nat Methods* (2011)). Copied below for the reviewer's convenience, Fig. 6 enables a direct comparison between fUS and fULM. The reviewer is indeed correct that fUS demonstrated robust sensitivity in identifying the activated brain regions when compared to fULM using conventional MB localization. Moreover, the activated regions detected by fULM using LOCA-ULM corresponds closely with the areas identified by the fUS (Fig. 6e). The resolution advantage of fULM over fUS is also clearly demonstrated in Fig. 6.

The following content has been added to the revised manuscript to reflect the new fUS results.

Page 17 Line 420 LOCA-ULM presents strong activation in the vessels within the barrel field of the primary somatosensory cortex (S1BF) and the ventral posterior medial nucleus (VPM)—regions corresponding to whisker stimulation. As illustrated in Fig. 6c, functional ultrasound (fUS) demonstrates

enhanced sensitivity in identifying the increased blood flow, detecting more extensive areas of activation compared to fULM using conventional localization. In contrast, the activated regions detected by fULM with LOCA-ULM corresponds closely with the areas identified by the fUS, while also offering a significant improvement in spatial resolution.

Page 32 Line 777 Then, fUS data acquisition was performed using a L22-14vX high frequency transducer connected to a Vantage 256 ultrasound system. fUS data were acquired using a 9-angle compounding plane-wave imaging sequence (step size of 1°) with a 40V transmit voltage. The transmit frequency was set to 15.625MHz with a transmitting PRF of 28.57 kHz and a post-compounding frame rate of 1,000 Hz. 250 frames of post-compounding IQ data were acquired per second, and a total of 180 acquisitions were acquired (total 3 minutes of acquisition). SVD-based spatiotemporal clutter filtering was used to suppress tissue clutter following the methods described in our previous fUS imaging study⁴⁸. Power Doppler images were generated by integrating 250 frames. The fUS activation maps were then created based on Pearson's product-moment correlation coefficient between the stimulation pattern and the Power Doppler signals for each pixel, as described by Macé, E. et al.³².

- Macé, E., Montaldo, G., Cohen, I. et al. *Functional ultrasound imaging of the brain*. *Nat Methods* 8, 662–664 (2011).

2. Regarding the Negative Correlations Outside Activated Regions: This is an interesting observation, and we can confirm that our results are consistent with the results published by Renaudin, N. et al., *Nature Methods*, (2022) (see Fig. 1e), where significant negative correlation was also presented in areas outside the activated regions. In fact, when displaying the correlation coefficient in fUS activation maps (e.g., Fig. 6c), one can also observe similar negative correlations. However, we agree with the reviewer that the fULM activation maps are in general noisier than fUS maps. This case represents a classic engineering tradeoff where a higher spatial resolution comes at the cost of lower SNR. While the exact physiological mechanism behind the negative correlations remains to be investigated, we do not believe that all the negative correlations are noise based on our results as well as the results published independently by Mickael Tanter's group (Renaudin, N. et al., *Nature Methods*, (2022)).

In response to the reviewer's request, we have also added the activated brain regions (i.e., pixels with positive correlation coefficient) identified by the LOCA-ULM fULM in Supplementary Fig 9.

Supplementary Fig 9. Regions with positive correlation identified by LOCA-ULM during functional ULM (fULM) imaging. The map highlights areas with increased blood flow correlating to whisker stimulation, with a color scale representing the Pearson's correlation coefficient.

Finally, we have noted in the main text the increased MB injection rate for fULM imaging experiment.

Page 17 Line 408 For the fULM experiment, we used an MB injection rate of 60 $\mu\text{L}/\text{min}$ to increase the MB concentration in the cerebrovasculature of the rat brain to increase fULM sensitivity.

- Renaudin, Noémi, et al. "Functional ultrasound localization microscopy reveals brain-wide neurovascular activity on a microscopic scale." *Nature methods* 19.8 (2022): 1004-1012.
- Liu, Thomas T. "Noise contributions to the fMRI signal: An overview." *NeuroImage* 143 (2016): 141-151.

R3.12) Methods line 626. The chicken embryo experiment is very interesting. It allows a direct comparison between the optical image (a ground truth) and the ULM image (Fig. 2.e). The description of this experiment is minimal in the methods. Since the imaging planes in optics and ultrasound are orthogonal (optics sees the surface, ultrasound cuts the medium), how do you place the ultrasound probe in the same plane as the optical image? Please provide a detailed description of this experiment.

Response We appreciate the reviewer's interest in the chicken embryo (CAM) experiment. We have expanded the methodological details in the revised manuscript.

Page 30 Line 735 CAM ultrasound imaging was performed using the Vantage 256 system (Verasonics Inc., Kirkland, WA, USA) system with a high-frequency linear array transducer (L35-16vX, Verasonics Inc., Kirkland, WA). The transducer was placed at the side of the plastic holder to image the CAM surface through a lateral acoustic window. Optical microscopy was conducted simultaneously using a Nikon SMZ800 stereomicroscope (Nikon, Tokyo, Japan) with a DS-Fi3 digital microscope camera (5.9-Mpixel CMOS image sensor, Nikon). The optical microscope was positioned above the weigh-boat and aligned to register the optical image with the ultrasound imaging plane. The initial positioning of the ultrasound-optical plane was determined using real-time B-mode imaging of native red blood cell scattering at a high transmit voltage (30V). Following this, the voltage was reduced to 6V, and the chicken embryo was injected with microbubbles in preparation for contrast-enhanced imaging. Ultrasound data were obtained by using a 9-angle compounding plane-wave imaging sequence (step size of 1°) with a center frequency of 20 MHz, pulse repetition frequency (PRF) of 40 kHz, and a post-compounding frame rate of 1,000 Hz. IQ data of 1600 frames per acquisition with a total of 20 acquisitions were generated (total 32 seconds of acquisition).

R3.13) Methods. The different experiments use different parameters, bubble concentrations and ultrasound parameters (1° steps or 2° steps, 5 or 9 angles, 15MHz and 20MHz, the voltage is only given in the fUS 6V) Can you summarise all the parameters in the supplementary table?

Response Following the reviewer's suggestion, we have modified Supplementary Table. 1 to summarize all the ultrasound parameters and MB concentrations used for different experiments.

	CAM	Rat Brain		
	Standard Imaging	Standard Imaging	fULM Imaging	fUS Imaging
In vivo acquisition parameters				
Transducer type	L35-16vX	L22-14vX	L22-14vX	L22-14vX
Center Frequency	20 MHz	15.625 MHz	15.625 MHz	15.625 MHz
Sampling Frequency	125 MHz	62.5 MHz	62.5 MHz	62.5 MHz
Wavelength	77 μm	98.56 μm	98.56 μm	98.56 μm
No. of compounding angles	9	5	5	9
Step size	1°	1°	2°	1°
Voltage	6V	6V	6V	40V
Concentration				
	70 μL Bolus injection	15, 20, 30, 40, 90 $\mu L/min$	60 $\mu L/min$	Contrast-free
Image resolution				
MB template pixel resolution	4.928 μm	9.856 μm	9.856 μm	
DECODE network input pixel resolution	9.856 μm	19.712 μm	19.712 μm	
DECODE network output pixel resolution	4.928 μm	9.856 μm	9.856 μm	

Minor questions

R3.14) Figure 1. The legend inside panel e (DECODE Network) is not visible and not explained in the caption or in the text.

Response We have enhanced the visibility of the legend by increasing its font size. Additionally, the DECODE network architecture is detailed in the Methods section under "DECODE Architecture." To improve cross-referencing, we have now modified the figure caption to explicitly direct readers to this section for a detailed explanation. The revised caption for Figure 1 is as follows:

Fig 1. Overview of the proposed LOCA-ULM MB localization pipeline. LOCA-ULM is a simulation-based supervised learning method using **MB templates** generated by Least-squares Generative Adversarial Network (LSGAN)²⁶ and **DECODE localization network**²⁴ (Methods – DECODE Architecture)

R3.15) Line 132. The acronym CAM is not specified in the text.

Response We have added the acronym as:

Page 6 Line 154 A representative simulated image using an LSGAN-generated MB signal is shown in Fig. 1b (LSGAN), which closely resembles real *in vivo* **chicken embryo chorioallantoic membrane** (CAM) images (Fig. 1d, Real image).

R3.16) line 284. “Our result suggests that the higher number of microvessels revealed by LOCA-ULM is indeed from localizations of real MBs instead of false ones”. This is a hypothesis, but it is difficult to verify.

Response We agree that the hypothesis is difficult to verify and revised the manuscript to remove this statement.

R3.17) Figure 6. Can you overlay the atlas to show that the activated regions are actually S1BF and VPM (a simple line might suffice)

Response In Fig. 6e, we have overlaid the atlas outline (corresponding to bregma: -3.3 mm) onto the fULM images. This overlay illustrates that the regions activated correspond accurately to the S1BF and VPM.

R3.18) Figure 6 Can you indicate the position of the imaged plane relative to the standard bregma point? (idem in the other figures)

Response We have indicated the position of the imaged plane in relation to the standard bregma reference point (bregma: -3.3 mm) in the legend of Figure 6. This information has also been incorporated into the Methods section of the main text. The position of the imaged plane for Fig. 3, 4, 5 have been also updated in the Figure legends.

Page 32 Line 774 Following the craniotomy, the target imaging plane including both S1BF and VPM at the bregma coordinate of -3.3 mm, was identified using 2D Power Doppler ultrasound imaging.

Fig 3. Comparison of LOCA-ULM and conventional localization to *in vivo* rat brain ultrasound data. **a** Power Doppler image generated by accumulating 2500 frames (a total of 2.5 seconds of acquisition, bregma: -4.4 mm) of rat brain ultrasound data.

Fig 4. Effect of different MB injection rates ($20 \mu\text{L}/\text{min}$, $30 \mu\text{L}/\text{min}$, $40 \mu\text{L}/\text{min}$) on LOCA-ULM and conventional localization for rat brain ULM imaging. **a-h** Each ULM image was generated by accumulating 25000 frames of ultrasound data (a total of 100 seconds of acquisition, bregma: -5.6 mm)

Fig 5. Comparison among LOCA-ULM, conventional ULM, Deep-ULM¹⁴, mSPCN²¹ in *in vivo* rat brain ultrasound data under high MB concentration (injection rate $40 \mu\text{L}/\text{min}$) **a** Each ULM image was generated by accumulating 25000 frames of ultrasound data (a total of 100 seconds of acquisition, bregma: -5.6 mm) for MB injection rate of $40 \mu\text{L}/\text{min}$.

Fig 6. LOCA-ULM increases MB signal sensitivity to blood flow during brain activation. **a, b** Schematic of the experimental setup for functional ultrasound (fUS) and functional ULM (fULM) brain imaging conducted in the coronal plane (bregma: -3.3 mm).

R3.19) The voltage is only 6V (in the fUS experiment). Higher voltages destroy the bubbles?

Response Our choice of 6V in the fULM experiment is based on two considerations. First, higher voltages pose a risk of disrupting the MBs near the cortical surface, and we used 6V to preserve these MBs. Secondly, increasing the voltage also amplifies the backscattering signal from the red blood cells. While these signals are less visible compared to MB signals, they can be mis-localized as MBs which manifests as noises in the final ULM image. Using an imaging voltage of 6V, we effectively enhance the signal from MBs while minimizing the interferences from the native red blood cells. This is essential for measuring fULM response which heavily relies on accurate MB counts.

R3.20) There is no mention of the ultrasound electronics and software in the methods (I assume a Verasonics?).

Response The reviewer is accurate. We have added the ultrasound system (Vantage 256 system) in the Methods.

Page 30 Line 735 CAM ultrasound imaging was performed using the Vantage 256 system (Verasonics Inc., Kirkland, WA, USA) with a high-frequency linear array transducer (L35-16vX, Verasonics Inc., Kirkland, WA).

Page 31 Line 763 All rat brain data were acquired using a high-frequency linear array transducer (L22-14vX Verasonics Inc., Kirkland, WA) connected to a Vantage 256 ultrasound system.

R3.21) Supplementary table. The 15.625Mhz of the L22-14vX probe is probably not the central frequency. It's probably the demodulation of the Verasonics electronics (it samples at 4x15.625Mhz). Check the central frequency in the probe calibration.

Response Thank you for highlighting the potential discrepancy. Upon review, we confirm that the default center frequency of the Verasonics L22-14vX probe is 18 MHz. However, for this study, we adjusted the center frequency to 15.625 MHz to improve SNR and better penetration. We also used 15.625 MHz to enable four samples per wavelength, given that the maximum sampling frequency of the Verasonics system is 62.5 MHz.

Reviewer #1 (Remarks to the Author):

The authors have addressed the reviewers' comments and revised the paper appropriately. The paper is now ready for publication.

Reviewer #2 (Remarks to the Author):

The authors have addressed all of my main concerns. As a minor suggestion, I would perhaps recommend reconsidering whether the first paragraph (line 41-48) is necessary and revisiting line 90-96, which could be shortened and clarified.

Reviewer #3 (Remarks to the Author):

The authors provided detailed and clear answers to all my questions. The corrected mss includes new experiments figures and clarifications that can help the reader to understand the interest and the scope of the proposed method. I recommend the publication of this article.

We thank all the reviewers for their positive and constructive comments. Reviewer 1 and Reviewer 3 did not have additional comments. We have addressed the minor comment provided by Reviewer 2 as follows,

Reviewer #2 (Remarks to the Author):

The authors have addressed all of my main concerns. As a minor suggestion, I would perhaps recommend reconsidering whether the first paragraph (line 41-48) is necessary and revisiting line 90-96, which could be shortened and clarified.

Response We have considered the suggestions from Reviewer #2 and made the following revisions.

(1) The reviewer suggested that the first paragraph is not necessary (line 42-48). We have shortened the first paragraph and integrated it into the subsequent paragraph, now reflected on Line 35-38 of the revised manuscript.

Page 2 Line 35 Single-molecule localization microscopy (SMLM) is an established super-resolution optical imaging technology that uses the stochastic blinking of fluorophore emissions within a dense sample^{1,2}. By localizing individual emissions and accumulating the localized positions, SMLM reconstructs a super-resolved image, offering an order-of-magnitude improvement in imaging spatial resolution³.

(2) The reviewer suggested to shorten and clarify Line 90-96. We have enhanced the clarity and conciseness of the writing, now reflected on Line 80-84 of the revised manuscript.

Page 4 Line 80 To account for these effects, a simulation framework was designed to model nonlinear wave propagation and MB responses, generating single-channel RF signals¹⁹. A 1D CNN was trained to deconvolve MB signals from raw RF data, enhancing the axial resolution of beamformed images tenfold over standard B-mode imaging. However, this technique requires precise parameter tuning and system characterization to generate realistic training datasets.